# The Aerosol Research Observation Station (AEROS)

Karin Ardon-Dryer[1], Mary C. Kelley[1], Xia Xueting[1*], and Yuval Dryer[1]

1. Department of Geosciences, Atmospheric Science Group, Texas Tech University, TX
* Now at Department of Statistics, Ohio State University, Columbus, OH, USA

*Correspondence to*: Karin Ardon-Dryer (Karin.ardon-dryer@ttu.edu)

**Abstract.** Information on atmospheric particles' concentration and sizes are important for environmental and human health reasons. Air quality monitor stations (AQMSs) for measuring Particulate Matter (PM) concentrations are found across the United States, but only three AQMSs measure $PM_{2.5}$ concentrations (mass of particles with an aerodynamic diameter of <2.5 µm) in the Southern High Plains of West Texas (area $\geq 1.8 \times 10^5$ km$^2$). This area is prone to many dust events (~21 per year),

yet no information is available on other PM sizes, total particle number concentration, or size distribution during these events. The Aerosol Research Observation Station (AEROS) was designed to continuously measure these particles' mass concentrations ($PM_1$, $PM_{2.5}$, $PM_4$, and $PM_{10}$) and number concentrations (0.25 – 35.15 µm) using three optical particle sensors (Grimm 11-D, OPS, and DustTrak) to better understand the impact of dust events on local air quality. The AEROS aerosol measurements unit features a temperature-controlled shed with a dedicated inlet and custom-built dryer for each of the three

aerosol instruments used. This article provides a description of AEROS as well as an intercomparison of the different instruments using laboratory and atmospheric particles. Instruments used in AEROS measured similar number concentration with an average difference of $2 \pm 3$ cm$^{-1}$ (OPS and Grimm 11-D using similar particle size ranges) and similar mass concentration, with an average difference of $8 \pm 3.6$ µg m$^{-3}$ for different PM sizes between the three instruments. Grimm 11-D and OPS had similar number concentration and size distribution, using similar particle size range, and similar $PM_{10}$

concentrations (mass of particles with an aerodynamic diameter of <10 µm). Overall Grimm 11-D and DustTrak had a good agreement in mass concentration, comparison using laboratory particles was better than that with atmospheric particles. Overall, DustTrak measured lower mass concentrations compared to Grimm 11-D for larger particle sizes, and higher mass concentrations for lower PM sizes. Measurement with AEROS can distinguish between various pollution events (natural vs anthropogenic) based on their mass concentration and size distribution which will help to improve knowledge of the air quality

in this region.

## 1. Introduction

Particulate matter (PM) comprises microscopic solid and liquid particles suspended in the atmosphere, which can be generated by anthropogenic or natural sources. PM is categorized by the size of the particle, with $PM_{10}$ representing a mass of particles with an aerodynamic diameter up to 10 µm. $PM_4$, $PM_{2.5,}$ and $PM_1$ representing a mass of particles with an aerodynamic diameter

of up to 4, 2.5, and 1 μm, respectively. In general, PM measurements define as measurements where 50% of the particles with the defined diameter (e.g., $PM_{2.5}$) will pass through a size-selective inlet. Smaller PMs can stay in the atmosphere for a long time and travel far from their source. PM in the atmosphere determines air quality levels and has been found to degrade human health (World Health Organization, 2016; Shiraiwa et al., 2017). The health impact is associated with particles smaller than $PM_{10}$, as particles ranging from 5 to 10 μm can settle in the upper respiratory system when inhaled, and smaller particles, such as $PM_{2.5}$, can penetrate deep into the lungs (Ling and van Eeden, 2009; Goudie, 2014). The latter has been identified as a leading contributor to the global burden of disease (Cohen et al., 2017; Lim et al., 2012).

In the United States (US), the Environmental Protection Agency (EPA) uses air quality monitoring stations (AQMSs) to monitor ambient $PM_{10}$ and $PM_{2.5}$ as hourly and daily average mass concentrations, but these stations generally have sparse geographic coverage, located in fixed sites (mainly in large population centres) and are lacking in smaller cities and underdeveloped regions. Additional monitoring networks provide information on $PM_{2.5}$ and $PM_{10}$ in the US, including the Interagency Monitoring of Protected Visual Environments (IMPROVE) network, which provides an additional 150 remote and rural sites nationwide, but the $PM_{2.5}$ and $PM_{10}$ samples are collected only every third day and provide only daily values (Prenni et al., 2019). The EPA Chemical Speciation Monitoring Network (CSN) provides information on $PM_{2.5}$ and the chemical composition of ambient fine particles across 150 US urban sites (Solomon et al., 2014; EPA, 2022). The Surface PARTiculate mAtter Network (SPARTAN) also provides information on $PM_{2.5}$ and $PM_{10}$ concentrations, but it has only two sites in the US and none in the south-central part of the country (Snider et al., 2015). Low-cost sensors, such as PurpleAir, are also increasingly used across the US, but their efficiency is still under investigation (Ardon-Dryer et al., 2020; Barkjohn et al., 2021). None of the monitoring units mentioned above provides information on total particle number concentrations or particle size distribution. The National Oceanic and Atmospheric Administration Earth System Research Laboratory (NOAA/ESRL) Federated Aerosol Network provides this information, but it is stretched very thin, with only a few units across the US (Andrews et al., 2019).

Most of these monitoring methods are not affordable, with prices ranging from $50,000 to $250,000, but newer methods based on optical particle sensors are becoming increasingly popular. These sensors rely on the principle of single-particle elastic light scattering following Mie scattering theory, which enables determining the size and number of particles within a unit volume of air (Masic et al., 2020). While some of these low-cost sensors (prices lower than $500) are gaining popularity, their efficiency and accuracy compared to reference sensors are still in doubt (Masic et al., 2020; Ardon-Dryer et al., 2020). Mid-price optical particle sensors ($10,000 − $20,000) have the advantage of a slightly more affordable price (than those of reference units) as well as better accuracy than the low-cost units. Among the advantages of these units, they can provide various types of measurement; for example, the Grimm 11-D (Grimm Aerosol Technik GmbH & Co. KG, Germany; Grimm 11-D, 2021) provides information on total particle number concentration and size distribution as well as information on mass concentrations of various PMs. Some of these units can provide information on multiple mass fractions of PM simultaneously,

which is an advantage to the gravimetric system which provides the mass concentration of only a single fraction (Masic et al., 2020). Several studies have found mid-price optical particle sensors to be comparable to high-priced reference units as long as the mid-price optical particle sensors undergo a regular (e.g., yearly) service and re-calibration (Viana et al., 2015; Jaafari et al., 2018; Vasilatou et al., 2021).

The Southern High Plains in West Texas hosts a few of the reference monitoring methods. The West Texas region (an area larger than $1.8 \times 10^5$ km$^2$) has only a few, widespread AQMSs operated by the Texas Commission on Environmental Quality (TCEQ) (TCEQ, 2021) which measure only PM$_{2.5}$ concentrations and provide no information on other PM sizes, total particle number concentrations, or size distribution. While the air quality in this region is considered good overall (Kelley et al., 2020), the region experiences many dust events (~21 per year) that reduce air quality (Kelley and Ardon-Dryer, 2021). Therefore, routine, and long-term measurements are required for comprehensive monitoring of diverse pollution events in this region, including dust events (Tong et al., 2012; Mahowald et al., 2014). Hence, there is a need to monitor particle mass concentrations (of various PM sizes) and size distribution to understand how they change under distinct metrological and pollution conditions. The Aerosol Research Observation Station (AEROS) was designed to address this need. This article provides information on each of its aerosol instruments and compares the units using standard particles in the laboratory as well as atmospheric measurements. Examples of aerosol measurements in various atmospheric conditions are presented to highlight AEROS's acuity in distinguishing between anthropogenic and natural pollution events.

## 2. Research Area and Measurement Station

### 2.1 Research Area

Measurements were taken in Lubbock, Texas, located in the Southern High Plains of West Texas (Fig. 1). This area is rural, flat, and approximately 1 km above sea level, with an urban area surrounded by extensive agriculture fields, including cotton (30% of national production) and cattle. It is a semi-arid environment with an average annual rainfall of 463 mm from 2000 through 2019, while the average annual rainfall for the same period in the US was 789 mm (Jaganmohan, 2021). The bare soil, low soil moisture, and strong winds typical of this region are important factors in dust formation (Stout, 1989). Several studies have found that this area is among the most prominent regions of dust events in the US (Orgill and Sehmel, 1976; Deane and Gutmann, 2003).

### 2.2. AEROS

AEROS was installed 9.8 m above the ground on the rooftop of the Electrical Engineering building at Texas Tech University (33°35'12.5"N 101°52'31.3"W; Fig. 1). AEROS's design followed World Meteorology Organization Global Atmosphere

Watch (WMO/GAW) aerosol measurement procedures, guidelines, and recommendations (WMO, 2016). It includes two units: an aerosol measurements unit and Harvard Impactor (HI) filter sampler unit.

95

The HI filter sampler unit has two setups with three HI units in each (Fig. 1). The HI units collect daily gravimetric $PM_{2.5}$ and $PM_{10}$ particles on filters substrates (Marple et al., 1987) in 24-hour cycles (midnight to midnight). The HIs are designed to sample particles of 2.5 µm and 10 µm at flow rates of 16.7 and 10.0 L min$^{-1}$, respectively, using impactor stages in series with polyurethane foam (PUF) impaction substrates (Lee et al., 2011). The 37-mm filters are pre- and post-weighed using an

100 electronic microbalance (XRP2U Microbalance) to provide gravimetric measurements. Filter are kept in the filter room which follows U.S. EPA (EPA, 1997) regulation conditions of temperature in the range of 20-23 °C and relative humidity in the range of 30-40%. To assure filter weight post sampling will not be impacted by hysteresis effect, filters post-aerosol collections are kept in a Dry-Keeper Auto-Desiccator Cabinet for 48 hours until weight. The filter sampler unit was fully operational only after September 2019 and therefore is not discussed in this article.

105

The aerosol measurements unit has been operational since March 14, 2019. It includes a shed that is temperature controlled by an air conditioning unit (Pioneer inverted WAS/WYS Series) that maintains a continuous temperature of 22 °C. Four rain-protected sampling inlet units are installed at 2.9 m from the rooftop floor ($1 \pm 0.01$ m from AEROS rooftop) to minimize influences from the surrounding area. Each rain-protected inlet unit collects total suspended particles and is connected to a

110 stainless steel tube (0.013 m diameter; 0.5 inches). Inside the station, each stainless steel inlet tube (from the outside) is connected to a custom-built in-line dryer unit that removes condensed-phase water from the collected particles (Fig. 1). Each dryer is 0.5 m long and contains a 0.013 m diameter metal wire mesh screen. A Swagelok reducer connects the dryers to a 0.0064 m diameter (1/4 inch) stainless steel tube. Conductive silicone tubes connect the small stainless steel tubes to the various instruments. Each inlet is connected to a different instrument, the flow in each inlet varies based on the instrument used (1.0

115 or 1.2 L min$^{-1}$). The average distance from the dryer to the instrument is about 0.24 m. Figure S1 provides a schematic design of the inlet to the instruments. There are no bends tubes in any part of the inlet tubes from the inlet to the instrument, and the entire sampling tube was kept to a minimum to minimize diffusion losses. Also, all the inlet tubes are aligned with the dryer and with the instrument to minimize particle loss. A calculation of the Reynolds number (Re) of each inlet and its instrument indicated that the aerosol flow in the inlets tubes is laminar (Re < 850). Calculation of particle loss in inlets (from rain protector

120 to instrument) was performed using the particle loss calculator (von der Weiden et al., 2009). Calculations were made for particles in the size range of 0.25 to 41 µm (based on particle size measured by instrument), using different particle types of different density and shape factors (based values in Table 1 in Ardon-Dryer et al., 2015). Particle loss was below 5% for particles of 0.25 µm and below 0.01% for particles in the size range of 1-2 µm.

125

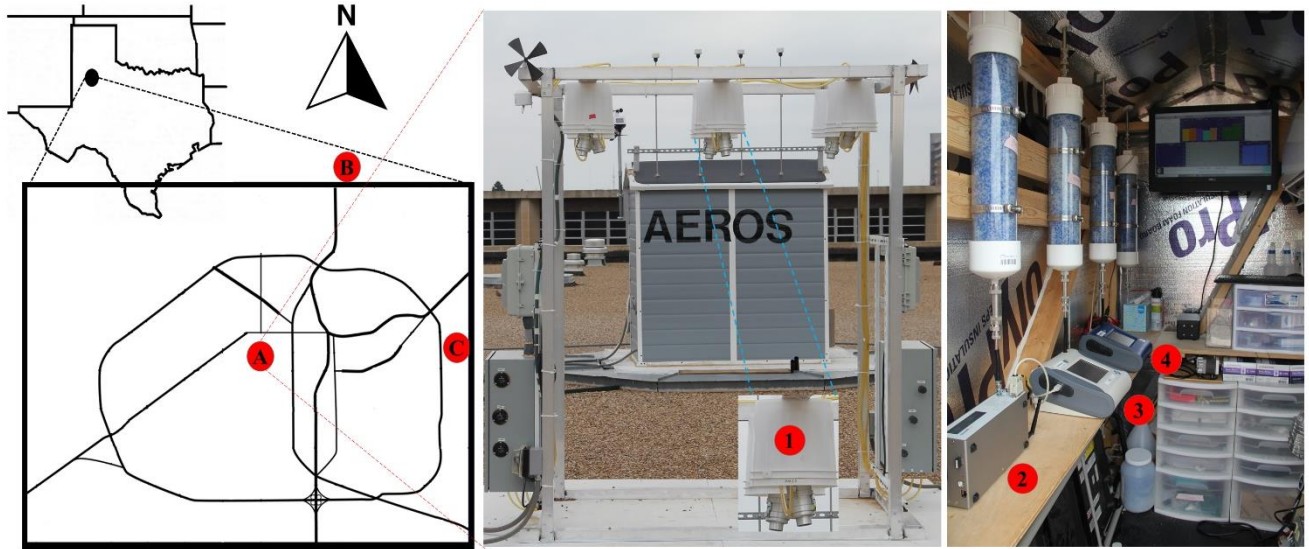

Figure 1. Location of AEROS (A) in the South High Plains of West Texas with locations of meteorological station (B) and TCEQ $PM_{2.5}$ station (C). The photos show the filter sampler unit with Harvard Impactor (HI;1) units and the aerosol measurements unit (outside and inside view with dryers and instruments 2-Grimm 11-D, 3-OPS, and 4-DustTrak).

### 2.2.1 Instruments used in the aerosol measurements unit

Each of the three inlets is connected to a separate aerosol instrument, and an additional inlet is kept available for aerosol collection using a filter holder (see Fig. 1). Three distinct particle instruments monitor PM concentrations, total particle number concentrations, and size distributions. The three instruments include TSI 3330 Optical Particle Sizer (OPS) (TSI, OPS3330 Shoreview, MN, USA), a DustTrak DRX aerosol monitor (TSI 8533EP, Shoreview, MN, USA), and Grimm 11-D system Portable Aerosol Spectrometer (Grimm Aerosol Technik GmbH & Co. KG, Germany). The three instruments are on a build shelf at the same height and at sufficient distance from one another to avoid interference (Fig. 1).

The OPS unit measures total particle number concentration as well as particle size distributions in 16 channels (bins) from 0.3 to 10 µm. It works on the principle of optical scattering from single particles. Particles are illuminated using a laser beam shaped to a thin sheath that is focused below the inlet nozzle. As particles pass through this light sheath, they scatter light in the form of pulses that are counted and sized simultaneously. The OPS time resolution is 1 min, with a flow rate is 1.0 L min$^{-1}$, which can reach a particle number concentration of up to 3,000 particle cm$^{-3}$ with a size resolution of < 5% at 0.5 µm and with measurements error of 0.001 cm$^{-3}$ (TSI, personal communication). There is an option to calculate total mass concentration for particles of up to 10 µm (representing $PM_{10}$). The OPS is calibrated by the manufacturer using different sizes of polystyrene latex sphere particles (PSL). In the operation of the OPS, the particle density is assumed as 1 g cm$^{-3}$, and no information on the reflective index is added, as there is very limited knowledge of the atmospheric particle chemical and mineralogical

composition in this region (Gill et al., 2000; 2009) and, therefore, no way to correctly capture the particles' density or refractive index, which are needed to convert the particle concentrations which are based on optical diameter to aerodynamic sizes. The OPS has been used previously in many laboratory settings (Ardon-Dryer et al., 2015; Yamada et al., 2015; Hsiao et al., 2016) and indoor experiments (Mølgaard et al., 2015; Maragkidou et al., 2018; Wang et al., 2020). Several studies that examined the performance of the OPS under diverse laboratory conditions have found it to be comparable with various reference units (Ardon-Dryer et al., 2015; Vasilatou et al., 2021). To the best of our knowledge, the OPS has not previously been used for atmospheric measurements or for monitoring atmospheric dust events.

The DustTrak DRX measures aerosol mass concentrations at various sizes ($PM_1$, $PM_{2.5}$, $PM_4$, and $PM_{10}$) at a time resolution of 1 min, using a flow rate of 1.0 L min$^{-1}$. Its detection ranges from 1 to 150,000 µg m$^{-3}$, with a mass resolution of 1 µg m$^{-3}$ (TSI Inc., 2019). Measurements are made with a diode laser wavelength of 655 nm (Wang et al., 2009). The DustTrak combines the photometric measurements of the group particles in the chamber with the optical sizing of single particles in the optical system and thus reports the concentration of various size fractions simultaneously. The unit is used with an external pump designed for continuous operation. The DustTrak is calibrated by the manufacturer using Arizona Road Dust/ISO 12103-1, and the default calibration factor ("Factory Cal") of 1.0 was used (TSI Inc., 2019). No information is provided by the manufacturer on the calculation or measurements error of the DustTrak. The DustTrak DRX (and previous versions) have been widely used in numerous studies (Holstius et al., 2014; Wang et al., 2020; Javed and Guo, 2021), mainly for monitoring outdoor PM due to its sensitivity to a diverse range of aerosols, fast response time, and high temporal resolution (Rivas et al., 2017). While some studies have reported high correlations of PM values between the DustTrak and a reference method (McNamara et al., 2011; Viana et al., 2015), others have found large differences between the two (Holstius et al., 2014; Javed and Guo, 2021). A better comparison can be achieved when relative humidity is taken into account with the use of a dryer (Javed and Guo, 2021).

The Grimm 11-D measures particle count and mass distribution by light scattering over the size range of 0.25 – 35.15 µm in 31 predefined size channels (bins). It provides measurements of total particle number concentrations, size distribution, and mass concentration (e.g., $PM_1$, $PM_{2.5}$, $PM_4$, and $PM_{10}$). Data are recorded at 1 min intervals (it is also possible to save data every 6 s). Particle mass concentration can reach up to 100 mg m$^{-3}$, while number concentration can reach up to 3,000 cm$^{-3}$. The Grimm 11-D tolerance ranges are ± 3% for particle concentration ≥ 500 cm$^{-3}$, and ±2 µg m$^{-3}$ (Grimm, personal communication). The sample volume flow is automatically regulated to the set point of 1.2 L min$^{-1}$. The air is drawn in via a radially symmetrical suction head and directed straight into an optical measuring cell with a diode laser wavelength of 655 nm (Peters et al., 2006). The signal from the scattered light is classified by size and count, and these counts are then converted to mass concentrations. These are made available through a Grimm proprietary algorithm, but the manufacturer does not share information about it, or the refractive index, density, and weighting factors used for the calculations. The Grimm 11-D is calibrated by the manufacturer using PSL particles according to ISO 21501-1, calibration factor ("Factory Cal") of 1.0 was

used (Grimm 11-D, 2020). Since the Grimm 11-D provide the concentration of particles for each bin size, calculations of size distributions for number ($dN/dlogD_p$) and volume ($dV/dlogD_p$) concentrations were performed from the instrument output using Matlab. The Grimm 11-D and previous versions have been used in various indoor (Mølgaard et al., 2015) and atmospheric studies (Mukherjee et al., 2017; Stavroulas et al., 2020; Masic et al., 2020), including under dusty conditions (Jaafari et al., 2018). Several studies examining the performance of the Grimm 11-D unit under diverse atmospheric and laboratory conditions have found it to be comparable to various reference units (Masic et al., 2020; Vasilatou et al., 2021). For example, Masic et al. (2020) found that it performed well under diverse atmospheric and pollution conditions; when equipped with a dryer, it performed at a level comparable to that of a reference unit (a Beta Attenuation monitor).

The three instruments used in AEROS (Grimm 11-D, OPS, and DustTrak) have been found to perform similarly to reference instruments (Viana et al., 2015; Masic et al., 2020; Vasilatou et al., 2021) and to one another (Crilley et al., 2018; Wang et al., 2020); some studies have even used them as reference instruments (Mølgaard et al., 2015; Crilley et al., 2018; 2020). The rationale for using these three instruments is the overlap in measurements between them. For example, similar PM sizes are measured by the DustTrak and Grimm 11-D, and total number concentration and size distribution (at least for the size range of 0.3-10.0) are measured by both the OPS and Grimm 11-D. The usage of these three different distinct instruments as part of AEROS aerosol measurements unit was planned to overcome times of common instrument problems, e.g., connection issues, broken units, or the need for repair. Both the Grimm 11-D and OPS are connected to a computer that saves their data, while the DustTrak data are saved on the instrument. Those data were downloaded and saved every week after the silica gel replacement, the 1-min values were then calculated using a MATLAB code to determine the values based on various time intervals (e.g., 10-min, hourly, and daily average values). All instrument time was synchronized and converted to local Central Standard Time (CST).

Each aerosol instrument is connected to a dedicated dryer to minimize the airflow passing through each dryer and to allow for longer use of each dryer (one-week duration). The dryers remove water from the particles by reducing the relative humidity from the surrounding air, relative humidity after the dryer is low ($24 \pm 0.5\%$). The instruments and station underwent standard maintenance operations each week, including replacing the used silica gel in each dryer with freshly baked ones, cleaning each inlet and tubing, and replacing paper filters in each instrument. In addition, each instrument was examined to verify that it counted 0 particles with a clean purge zero count filter, which enabled testing for leakage. Additional zero offset calibrations were performed on the DustTrak, based on the manufacturer's advice. When no particles were detected, the freshly baked dryer was connected to each instrument with a clean filter at the inlet, and measurements of particles were performed to verify the dryer background particle level (PM, size distribution, and total number concentration). These background values were subsequently subtracted from the instruments' atmospheric measurements. The contribution of particles due to the use of the dryers was minimal; for example, the $PM_{10}$ particle mass concentration was $0.3 \pm 0.16$ µg m$^{-3}$ (average $\pm$ standard deviation, SD values), while the number concentration of particles in the size range of 253 - 298 nm was $15.4 \pm 8.9$ cm$^{-3}$.

## 2.3. Measurements of PM$_{2.5}$ concentration from TCEQ

The only reference AQMS unit in this region belongs to the TCEQ (TCEQ, 2021). This unit located 8.2 km from AEROS (Fig. 1), measures only hourly PM$_{2.5}$ concentrations (local conditions at local CST) using a Met One BAM-1022 Beta Attenuation Mass monitor unit. The BAM-1022 measures PM$_{2.5}$ concentrations ranging from -15 to 10,000 µg m$^{-3}$ with a resolution of 0.1 µg m$^{-3}$ and a precision of < 2.4 µg m$^{-3}$ per hour. Additional information on this unit can be found in Kelley et al. (2020).

## 2.4. Comparison of aerosol instrumentation under laboratory and atmospheric conditions

A comparison of the three aerosol instruments was performed using known particles under controlled laboratory conditions as well as under atmospheric conditions.

### 2.4.1. Comparison of aerosol instrumentation using known particles in the lab

Although the three instruments were received from the manufacturer after factory calibration, we performed calibration tests of the OPS and Grimm 11-D using three monodisperse polystyrene sphere particles (0.25, 0.5, and 0.95 µm) to verify their performance in identifying particle size at the corrected size bins. The PSL particles were wet generated using a Brechtel Manufacturing, Inc. (BMI) 9200 Aerosol Generator (BMI, 2022). The atomized particles entered integrated in-line dryers where they evaporated, leaving anhydrous crystalline particles before reaching OPS and Grimm 11-D.

A laboratory comparison was performed using an experimental setup designed specifically for this comparison (Fig. 2). For this comparison, Arizona Test Dust (ATD) particles (Nominal 0 - 3 mm, Powder Technology Inc. MN, US) with 100 µm Bronze Beads (TSI 3400) were generated using a 3D printed dust generator (PRinted FluidIZed bed gEnerator- 3D dust generator, PRIZE; Roesch et al., 2017). The dry dust particles were suspended in the dry generator using a 4 L min$^{-1}$ nitrogen flow. The particles were then measured by each of the three instruments, and any excess flow not drawn into the instruments was filtered and vented to a hood. A Brechtel Y-shaped flow splitter was used to split the flow, and conductive silicone tubing carried the particles between all components to minimize particle loss to the tubing by electrostatic forces.

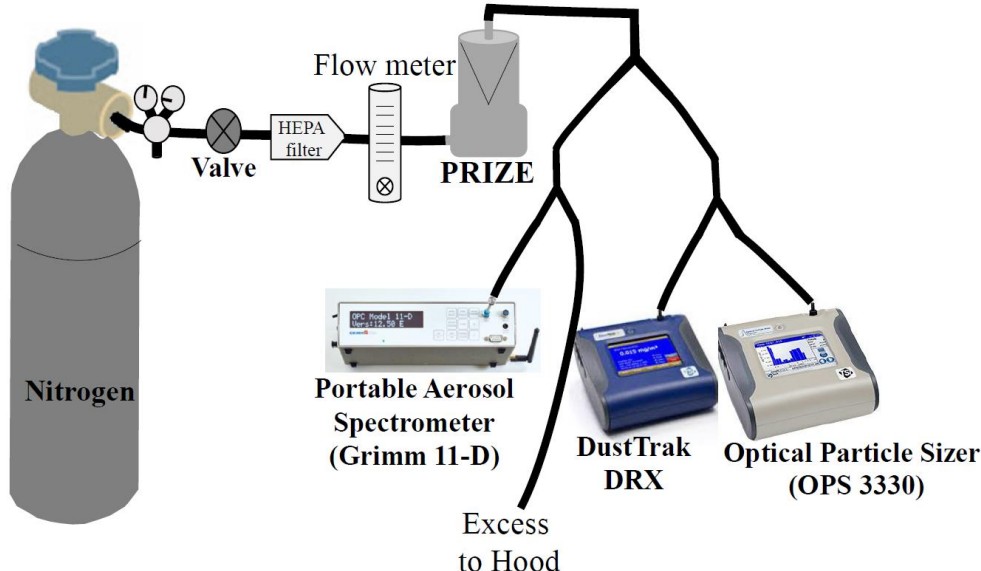

Figure 2. Experimental setup: Arizona Test Dust (ATD) particles were generated using PRinted FluidIZed bed gEnerator- 3D
dust generator (PRIZE) and measured by the various instruments (DustTrak, OPS, and Grimm 11-D).

### 2.4.2. Comparison of aerosol instrumentation using atmospheric particles

Two types of atmospheric measurements were performed using the three instruments. In the first, a comparison of the aerosol
instruments in AEROS was performed for 78 days from mid-March to the end of May 2019. In the second comparison, which
took place in the same period, aerosol measurements by AEROS were compared to measurements taken at ground level and
outside the station shed (on the rooftop).

To evaluate the similarities and differences of the three instruments (or locations), a set of calculations and comparisons was
performed using MATLAB and Excel. The evaluation and comparisons were based on R-squared ($R^2$), root-mean-square error
(RMSE), and mean absolute error (MAE) values as well as the best fit information (including the slope and intercept), and
Pearson correlation coefficient based on linear regression (standard least-squares linear regression). Additional evaluation
based on orthogonal distance regression was made using R. After the comparisons were performed, additional measurements
of different meteorological and atmospheric conditions were made to observe the behavior of AEROS and examine its ability
to observe diverse pollution conditions and to distinguish between natural (e.g., dust) and anthropogenic (e.g., haze) pollution.

### 2.5. Meteorological measurements

Meteorological information, such as 5-min to hourly ambient temperature, relative humidity, wind speed, direction, and gust
as well as visibility, pressure, and precipitation were retrieved from the local National Weather Service (NWS) Automated

Surface Observation System (ASOS), available via the METeorological Aerodrome Reports (METARs) station located ~9.8 km northeast of AEROS (33° 39' 48.96" N, 101° 49' 22.8" W, Fig. 1). The data were retrieved from March to May 2019, and all times were converted to CST. Observations of meteorological conditions (e.g., thunderstorms, rain, haze, and dust) were
retrieved for that period using the "Present Weather Code", which is provided in the METAR.

## 3. Results and Discussion

### 3.1. Laboratory intercomparison of aerosol instrumentation using known particles

Analysis of OPS and Grimm 11-D using PSL particles was performed to identify if the instrument can detect particles at the correct sizes. Three different PSL sizes were examined, these PSL had nominal sizes of 0.25, 0.5, and 0.95 µm with a size
range of 0.24-0.26, 0.48-0.52, and 0.93-0.97 nm, respectively. The results of the PSL test can be found in Fig. S2, on average 16 measurements were taken for each size and instrument. Overall, OPS and Grimm 11-D identified particles of similar sizes and concentrations. Both instruments, when examining 0.25 µm particles (Fig. S2A), had the highest concentration at the smallest (first) bin size, Grimm 11-D identified the 0.25 µm PSL at a bin size of 0.253 to 0.298 µm, while OPS detect the highest concentration at a bin size of 0.3 to 0.374 µm. When 0.5 µm PSL particles were examined (Fig. S2B), both units
identified monodisperse distribution with a narrow maximum at the expected size range. The OPS identified the PSL at size range (bins) of 0.465 to 0.579 µm while Grimm 11-D identified most of the particles in two bins of 0.414 to 0.488 µm and 0.488 to 0.576 µm, the particles examined were in the range of 0.48-0.52 µm, and therefore identified in the correct detected sizes of Grimm 11-D. For the 0.95 µm particles (Fig. S2C), both instruments behave similarly and had bimodal distribution with two maxima, one at the smallest bin and another one at larger particle size. We suspected that high concentrations of
small particles detected in this PSL solution were due to an artifact caused by surfactant used in the PSL solution. The surfactant is added by the manufacturer to help keep the spheres PSL from clumping together during storage, but often can produce a tail of small particles. OPS identified the 0.95 µm particles in size bins of 0.897-1.117 µm, while Grimm 11-D identified most of the particles in bin size of 0.679 to 0.8 µm, much lower than the PSL size range. More recently, when only Grimm 11-D was used (Fig. S2D) while using a new solution of 0.95 µm PSL particles, Grimm 11-D identified most of the particles in two bins
0.679 to 0.8 and 0.8 to 0.943 µm, the latter was in the PSL size range yet slightly lower than size expected. The detection of particles of that size range (~ 1 µm) at smaller sizes was observed in previous studies that used Grimm 11-D, yet it seems as if this size was in the detected size range according to ISO 21501-4 (Vasilatou et al., 2021). The behavior of the OPS came as no surprise as it was similar to previous studies that used size-selected ammonium sulfate particles (Ardon-Dryer et al., 2015).

Arizona Test Dust particles were generated and measured by each instrument every minute for 30 min. A comparison of total particle number concentration and size distribution was made between the OPS and the Grimm 11-D, while a comparison of PM was performed between the DustTrak and Grimm 11-D. Overall, similar measurements were found between the various instruments as shown in Fig. 3. Full information on the statistics based on liner regression of each comparison including $R^2$,

RMSE, and MAE, slope, intercepts, the number of parallel measurements, Pearson correlation coefficient value as well as slop

and intercepts based on orthogonal distance regression can be found in Table S1. The OPS and Grimm 11-D had a similar

particle size distribution in most of the overlapping particle sizes, mainly for particle sizes ranging from 0.8 µm to 9 µm. For

small particle sizes (<0.8 µm), however, the Grimm 11-D measured a higher particle number concentration than the OPS (Fig.

3A). Similar values of total particle number concentration were measured by the OPS and Grimm 11-D when similar particle

size ranges were used (0.3-10 µm; Fig. 3B). A high $R^2$ value ($R^2 = 0.97$) was measured during this experiment, and no statistical

difference (based on one-way ANOVA) was detected between the two units. A comparison of the DustTrak and Grimm 11-D

was performed using various PM sizes (Table S1 and Fig. 3C). Overall, both instruments measured similar PM concentrations,

but the Grimm 11-D measured higher mass concentrations for the larger particle sizes ($PM_{10}$, $PM_4$, $PM_{2.5}$), while the DustTrak

measured higher mass concentrations than the Grimm 11-D for $PM_1$. The $R^2$ for the PM concentration comparison was high

(range: 0.85 - 1.0, see Table S1), and there was no statistical difference between the measurements of these instruments based

on one-way ANOVA.

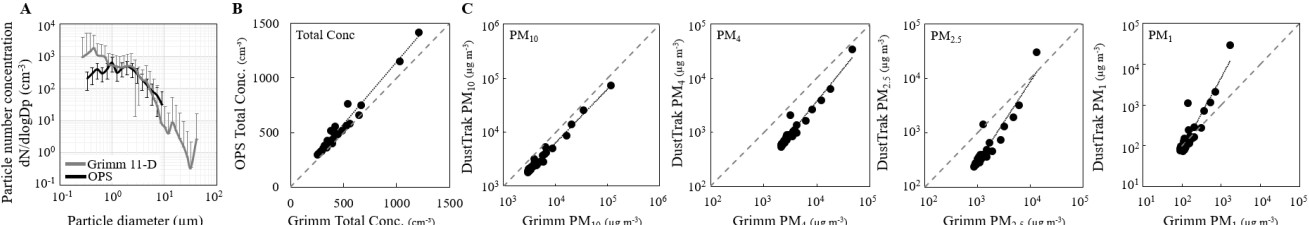

Figure 3. Comparison of the particle size distribution for optical diameter between the OPS and Grimm 11-D (A) total number

concentration (B), and comparison of PM concentration between the DustTrak and Grimm 11-D for $PM_{10}$, $PM_4$, $PM_{2.5}$, and

$PM_1$ (C) using Arizona Test Dust particles. Error bars represent SD values for measurement duration.

**3.2. Intercomparison of aerosol instruments using atmospheric particles**

A comparison of atmospheric measurements was performed using hourly average values measured from mid-March to the end

of May 2019 (a total of 78 days). During this period, PM and total number concentration varied, as shown in Fig. 4. The hourly

$PM_{2.5}$ values ranged from < 1 µg m$^{-3}$ to more than 300 µg m$^{-3}$, while the total number concentration ranged from 0.5 to 220

cm$^{-3}$. The time comparison in Fig. 4 shows that, while two instruments (OPS and Grimm 11-D) measured similar total number

concentration values, the three instruments that measured $PM_{2.5}$ values (Grimm 11-D, DustTrak, and TCEQ) had large

variabilities in their PM values. The Grimm 11-D measured higher $PM_{2.5}$ values on some days while on others, the DustTrak

measured higher $PM_{2.5}$ concentrations. For that difference, a full comparison was performed between all the instruments for

diverse PM sizes as well as for the total number concentration (Fig. 5). Additional information of each composition including

averaged and SD, median, mode, 10th, and 90th percentile values can be found in Table S2. It should be noted that during the

examined period the DustTrak reported no jumps in PM concentrations or negative or 0 values under low PM concentrations (as presented in Rivas et al., 2017), perhaps due to the weekly calibration (zero offset).

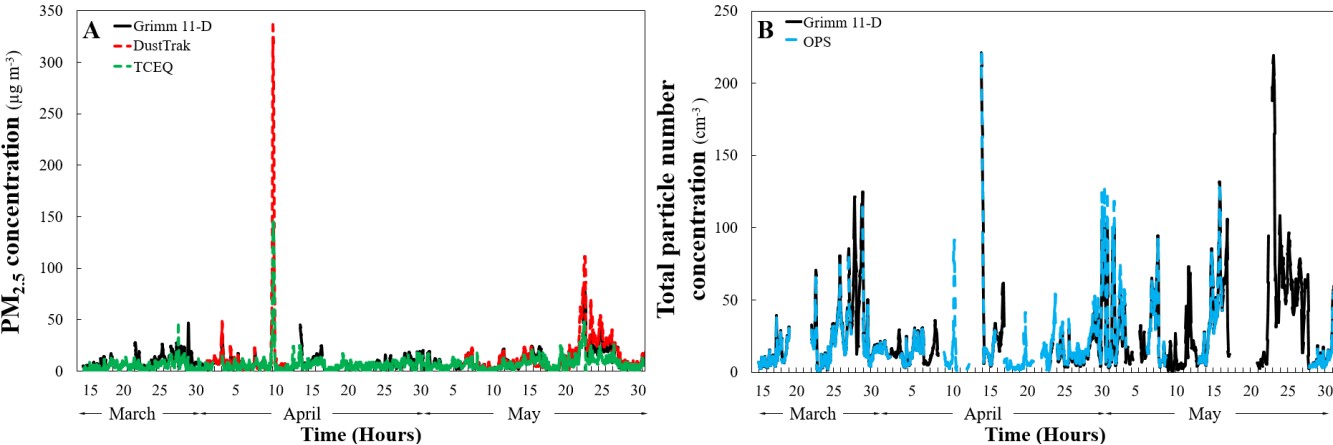

Figure 4. (A) Hourly values of PM$_{2.5}$ from the Grimm 11-D (black), DustTrak (red), and TCEQ station (green) and (B) total number concentrations from the Grimm 11-D (black) and OPS (light blue) as measured during March-May 2019.


A comparison of atmospheric measurements was performed for PM$_{10}$ between the Grimm 11-D and OPS (Fig. 5A). This comparison, which had 867 hours of parallel measurements, returned a high $R^2$ value (0.95) and low RSME and MAE values (3.3 and 2.1 μg m$^{-3}$, respectively). When the PM$_{10}$ values from the OPS were compared to those of the DustTrak (Fig. 5B), the comparison had a lower $R^2$ value (0.79) and higher RSME and MAE values (24.3 and 8.0 μg m$^{-3}$, respectively). Although this

comparison was low, previous studies have shown that the OPS and DustTrak measure similar PM$_{10}$ values, under laboratory conditions (Wang et al., 2020).

The PM concentration for sizes PM$_{10}$, PM$_4$, PM$_{2.5}$, and PM$_1$ were compared between the Grimm 11-D and DustTrak; there were 671 parallel hours. The $R^2$ values ranged from 0.63 (for PM$_{10}$; Fig. 5C) to 0.86 (for PM$_{2.5}$; Fig. 5D), while the RSME

values ranged from 5.3 μg m$^{-3}$ to 10.6 μg m$^{-3}$, and the MAE values ranged from 3.3 μg m$^{-3}$ to 6.6 μg m$^{-3}$. On average, the Grimm 11-D measured higher PM$_{10}$ and PM$_4$ values (9.3 ± 19.1 and 2.8 ± 8.4 μg m$^{-3}$, respectively) than the DustTrak, similar to the results of Javed and Guo (2021), who found that the DustTrak measured lower mass concentrations at larger particle sizes. For PM$_{2.5}$ and PM$_1$, however, the DustTrak measured higher values on average (2.4 ± 6.5 and 5.3 ± 8.2 μg m$^{-3}$, respectively) than the Grimm 11-D. These findings are similar to those of Holstius et al. (2014), who compared the DustTrak

and a Grimm unit and recorded higher PM$_{2.5}$ values from the DustTrak than from the Grimm, perhaps because the DustTrak overestimated the concentration of PM$_{2.5}$ (Javed and Guo, 2021).

In a comparison of PM$_{2.5}$ hourly values between the Grimm 11-D and DustTrak to the local TCEQ station (Figs. 5E, 5F) the AEROS instruments (Grimm 11-D and DustTrak) measured higher PM$_{2.5}$ values (with averages of 3.5 ± 5.5 and 6.1 ± 15.1 μg

m$^{-3}$, respectively) than those measured by the TCEQ. When PM$_{2.5}$ values from the TCEQ were compared with those measured by the DustTrak, the comparison had a high R$^2$ value (0.8) and low RSME and MAE values (4.8 and 3.3 µg m$^{-3}$, respectively). A lower R$^2$ value (0.55) and RSME and MAE values (3.5 and 2.5 µg m$^{-3}$, respectively) were measured when the TCEQ values were compared with those of the Grimm 11-D. Although the overall R$^2$ values were high and the RSME and MAE values were low overall, there were differences between the units. The differing PM$_{2.5}$ values between the TCEQ and the Grimm 11-D and DustTrak could be attributed to two causes. First, the TCEQ unit is not located near AEROS but ~8.2 km away meaning it was most likely exposed to slightly different conditions (e.g., due to its location near an agriculture field, while AEROS is located on campus in an urban setting) and therefore had different particle mass concentrations. Second, several of the TCEQ PM$_{2.5}$ values were below zero (down to -8 µg m$^{-3}$), and the TCEQ zero setting is below 0 µg m$^{-3}$, which could impact the comparison by lowering the overall TCEQ values.

A comparison of total particle number concentration between the OPS and Grimm 11-D for particles 0.3 µm to 10 µm yielded a high R$^2$ value (0.98) and low RSME and MAE values (3.5 and 2.5 cm$^{-3}$, respectively), with a slope of 1.0 (Fig. 5I) emphasizing the compatibility of the two units. It should be noted that although overall these two instruments show high comparability a close look at the distribution of the total concentration shows a difference between the OPS and Grimm 11-D over some periods. A comparison was performed between the units based on different periods, where each period represents the time between silica gel replacement and filter change in instruments (see Fig. S3). For two out of the nine periods (for unknown reasons) OPS measured much higher number concentration values compared to Grimm 11-D, leading to much higher difference values between the two units (Fig. S3B) and therefore shift of the 1:1 line (Fig. 5I).

Overall, the OPS and Grimm 11-D are more comparable based on their total number concentration and PM$_{10}$ values, but the Grimm 11-D and DustTrak had high comparison values (relatively high R$^2$ values) for the diverse PM sizes, so the difference was not consistent. Larger PM sizes (PM$_{10}$ and PM$_4$) were higher in the Grimm 11-D than in the DustTrak, while smaller PM sizes (PM$_{2.5}$ and PM$_1$) were higher in the DustTrak than in Grimm 11-D. Some of these differences in mass concentration in the atmospheric measurements could be attributed to slight changes in the method used by each instrument for particle detection. For example, according to Wang et al. (2020), the OPS uses a more focused laser beam and a nozzle with a smaller inner diameter to sample particles compared to the one used in the DustTrak, while the DustTrak single scattering measurement has a larger minimum detectable size (~0.5 µm) yields more coincidence errors than the OPS. Another factor lay with the fact that the instruments are calibrated by the manufacturer using different particle types, both OPS and Grimm 11-D calibrated using PSL particles while the DustTrak is calibrated with Arizona Road Dust. Calibration using different particle types could cause different detection or reading. Previous studies indicated that optical responses of different particles may vary significantly, depending on the particles type or the pollution level (McNamara et al., 2011; Sousan et al., 2016; Masic et al., 2020). For example, irregular particles, like dust particles, will scatter more light which may overestimate the optical diameter of the particles (Chien et al., 2016). According to Zhang et al. (2018), the relationship between PM mass concentration and

light scattering is strongly dependent on particle size and, to a lesser extent, on PM composition. Atmospheric particles, as the one used in this comparison, contain different types of particles which will be varied by their refractive indexes, densities, and shapes leading to slightly different interpretations by each of the instruments and to different readings (Cheng et al.,2010). Since there is very limited information about the atmospheric particle chemical and mineralogical composition in this region no correction (e.g., different refractive indexes, densities values) could be made, and instruments were used as default from the manufacturer with manufacture correction factors.

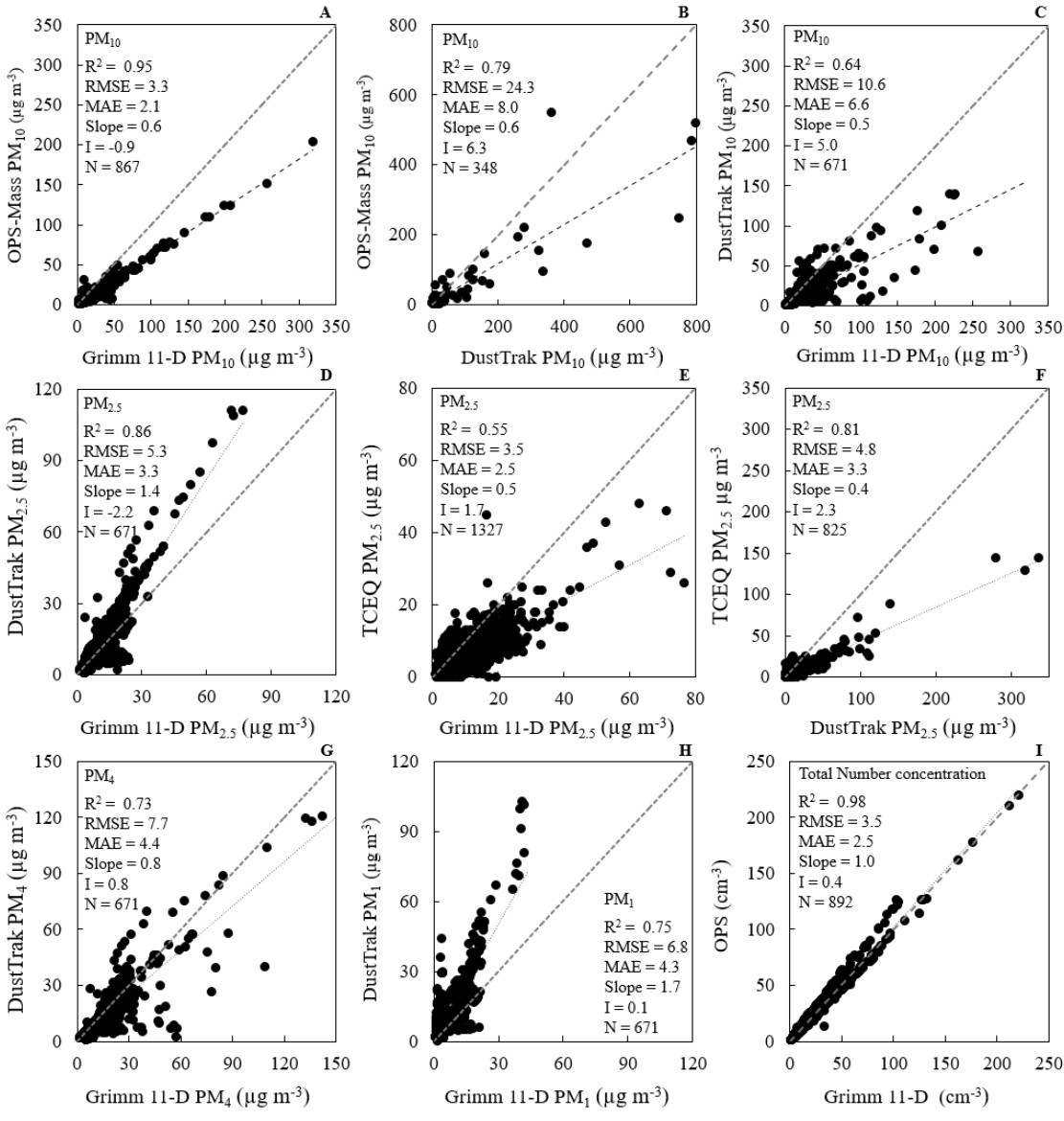

Figure 5. Instrument comparison based on linear regression, comparison of hourly PM, and total particle number concentration values as measured by the Grimm 11-D, OPS, DustTrak, and TCEQ. Dashed gray lines represent a 1:1 line. The statistics of

each case include the $R^2$, RMSE, and MAE, as well as the slope, intercepts (I), and N, which represent the number of parallel measurement points. Shown are comparisons of the Grimm 11-D and OPS (A) and Grimm 11-D and DustTrak (B) for $PM_{10}$

and between the OPS and DustTrak for $PM_{10}$ (C). The Grimm 11-D and DustTrak (A) and Grimm 11-D and TCEQ (B) for $PM_{2.5}$, and between TCEQ and DustTrak for $PM_{2.5}$ (E). Comparison between the Grimm 11-D and DustTrak for $PM_4$ (G) and $PM_1$ (H), and between Grimm 11-D and OPS for total particle number concentration (I). Additional statistics for each comparison can be found in Table S2.

### 3.3. Comparison of particle concentration based on different locations

A comparison of particle concentration (mass and number) based on instrument location was performed (using identical rental units). For this comparison, one Grimm 11-D unit was in AEROS, while the second (rental) unit was located outside the shed on the rooftop floor. One DustTrak and one OPS unit were kept in AEROS, while two other (rental) units were located on the ground floor. Each measurement in each location was taken every 1min for 1hour. The instruments in AEROS used the sampling design and inlet length described in Section 2.2 and shown in Fig. 1, while the units at the two other locations (rooftop

and ground floor) were used as is, without a dryer or inlet. These comparisons were taken under atmospheric conditions with a temperature of $26 \pm 5.4$ °C and relative humidity of $48.9 \pm 16.7$ % (as measured by the NWS station). A comparison of each instrument pair (near each other) showed that both units measured similar overall concentrations (number and mass, data not shown).

Overall, similar particle concentrations were found at all three locations (Fig. 6). The average particle size distribution measured in AEROS, when compared to those taken on the rooftop floor using the Grimm 11-D (Fig. 6A) showed similar number concentrations for all particle sizes. For the comparison between measured in AEROS and the ground floor using OPS (Fig. 6B), we found higher particle number concentration in size range of 0.3 to 2 µm (with difference up to 350 cm$^{-3}$ for 0.3 µm) at the ground level. The measurements at the ground floor were higher most likely due to people walking near the

instruments and kicking particles from the sidewalk, and the fact the ground sampling location was near a parking lot that was active during the sampling period. Although higher number concentrations were measured at the ground, the comparison between the two OPS measurements (in the AEROS shed and on the ground floor) had a high $R^2$ value (0.99) and low RMSE and MAE values (0.8 and 0.6 cm$^{-3}$, respectively). The difference between the two Grimm 11-D measurements (in the AEROS shed and on the rooftop floor) also had a good comparison, with a high $R^2$ value (0.99) but with slightly higher RMSE and

MAE values (7.7 and 3.4 cm$^{-3}$, respectively).

Similar measurements were obtained for PM concentration using the Grimm 11-D and DustTrak in different locations. The PM concentration measured using the Grimm 11-D in the AEROS shed were slightly higher (with an average of $2.3 \pm 1.3$ µg m$^{-3}$ for all PM sizes) than the measurements taken on the rooftop floor (Fig. 6C), while the measurements with the DustTrak

at ground level were also slightly higher (an average of $1.3 \pm 1.1$ µg m$^{-3}$ for all PM sizes) than those measured in the AEROS

shed (Fig. 6D). Although there were differences in PM concentrations, these were relatively small (1.3 - 2.3 µg m⁻³) and within
the range of difference found between the two instruments when they were measured at the same location and time. In addition,
in both cases, the RMSE and MAE were relatively low ($\leq 1.8$ and $1.2$ µg m⁻³, respectively). There was no statistical difference
(based on one-way ANOVA) between the measurements from these locations (in the AEROS shed vs. the rooftop floor or the
ground floor). Overall, this comparison showed that measurements using AEROS (with the current setup in the shed) reflect
measurements at ground level, at least for the condition tested. It is possible to assume that different meteorological and
atmospheric conditions would cause some differences.

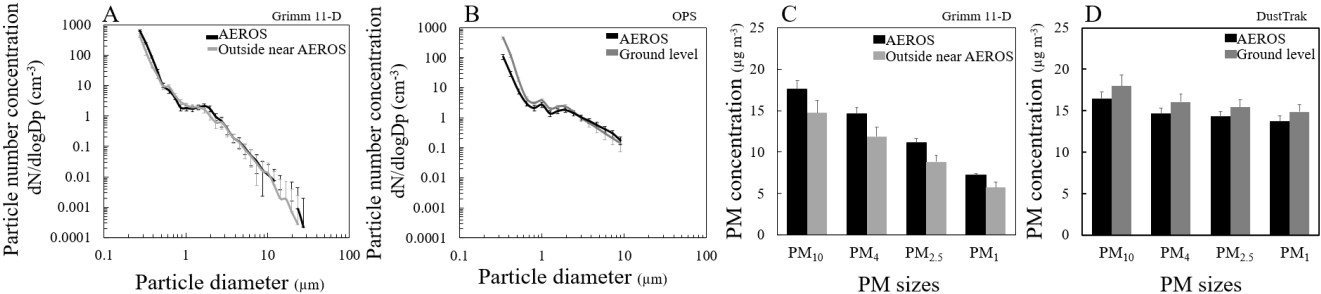

Figure 6. Comparison of measurements taken in the AEROS shed with measurements taken on the ground level or the rooftop
floor outside the AEROS shed. (A) Particle size distribution (optical diameter) measured by the Grimm 11-D unit in the
AEROS shed (black) and outside on the rooftop floor (light gray). (B) Particle size distribution (optical diameter) measured
by the OPS in AEROS (black) and on the ground floor (dark gray). (C) PM concentration at various sizes as measured by the
Grimm 11-D unit in AEROS (black) and outside AEROS on the rooftop floor (gray). (D) PM concentration at various sizes as
measured by DustTrak in AEROS (black) and on the ground floor (dark gray). Error bars represent SD values of the measured
period.

### 3.4. Observation and identification of different pollution events (anthropogenic vs. natural)

Observations using AEROS's aerosol instruments were used to distinguish between different pollution events attributed to
anthropogenic causes (haze) or natural causes (dust events). Ideally, the identification of particle chemistry confirms the type
of particles, but that was impossible at the time of the measurements, so observations of particle concentrations (total number
and mass concentrations) and particle size distribution were used to distinguish between these different events. It is expected
that pollution events will have high emissions of particles with a high particle concentration, as an anthropogenic event has
more small particles than a natural event (e.g., dust), which has larger particles (Kulkarni et al., 2011).

Observations of anthropogenic and natural events were made on March 28 - 30, 2019, when two haze events and one dust
event were captured. Figure 7 presents the total number concentrations, PM mass concentrations, and size distribution at these
times. During the morning hours of March 28, the local NWS reported a haze event. The visibility (based on measurements
taken from the meteorological station) decreased from 16 to 8 km (from 5:00 to 10:00). At 10:00, the hourly average value

based on total particle number concentration was 122.5 ± 14.1 cm$^{-3}$ (Fig. 7A), and the hourly PM concentration at the same time did not exceed 45 µg m$^{-3}$ (PM$_{10}$ was 44 ± 5.9 µg m$^{-3}$, PM$_{2.5}$ was 27 ± 1.4 µg m$^{-3}$, and PM$_1$ was 23.8 ± 1.2 µg m$^{-3}$; Fig. 7B).

The size distribution at the same time showed a very high number concentration of small particles <1µm (more than 10$^3$ cm$^{-3}$ for particles ranging from 0.25 - 0.3 µm; Fig. 7C). Haze was reported again the next morning beginning at 5:00, and the visibility from 10:00 to 11:00 dropped from 16 to 8 km. The total number concentration (hourly average) at that time was 126 ± 13.8 cm$^{-3}$ (Fig. 7A). The PM hourly concentrations did not exceed 30 µg m$^{-3}$ (the hourly PM$_{10}$ was 29.6 ± 3 µg m$^{-3}$, PM$_{2.5}$ was 23.0 ± 1.6 µg m$^{-3}$ and PM$_1$ of 20.8 ± 1.5 µg m$^{-3}$; Fig. 7B), while as observed the previous day, the size distribution showed

very high number concentrations of small particles of <1µm (Fig. 7C). The two haze events had relatively similar concentrations when the observation was made as a function of volume size distributions (Fig. 7D). These two haze events had lower (by an order of magnitude) particle mass and total number concentrations compared to several large haze events measured in China (Gou et al., 2014; Wang et al., 2014; Li et al., 2019). Some of the differences in particle concentration in the haze events measured here compared to those measured in China may be attributed to the different particle sizes used. The

particle size range used in Gou et al. (2014) was smaller (from 10 nm to 0.6 µm) than the one used in this work (particles ≥ 0.25 µm were detected). Using similar particle sizes, Wang et al. (2014) still measured higher particle number concentrations than the two presented here, but the haze event in their work had a higher magnitude than the one measured in this work. It is possible to assume that since measurements taken in this region which has much smaller cities compared to those measured in China, therefore there will be differences in the emissions rate and type which will attribute to the differences of number and

mass concentrations observed here compared to those from China.

On March 30 at midnight, a dust event (blowing dust) was reported by the local NWS station (reports observed between 0:35 to 0:45). The wind speed reached 12 m s$^{-1}$, wind gusts of 17 m s$^{-1}$ were reported, and the visibility dropped to 9.6 km. During that hour, lower total number concentrations were measured (28.3 ± 2.3 cm$^{-3}$) than those measured in the two haze events

mentioned above. Higher PM concentrations were reported during the hour with the dust event, with hourly values of 319.3 ± 192.2, 46.7 ± 29.9, and 6.6 ± 3.5 µg m$^{-3}$ for PM$_{10}$, PM$_{2.5}$, and PM$_1$, respectively (Fig. 7B). This dust event had much lower PM concentrations than those measured in Saudi Arabia (Alghamdi et al., 2015), Israel (Ardon-Dryer and Levin, 2014), Crete (Polymenakou et al., 2008), and other locations in the US, such as Arizona (Hyde et al., 2018). During the dust event, higher number, and volume concentrations of larger particles (>0.8µm) were observed (Fig. 7C, 7D). The size distribution of the

particles had a bimodal distribution, with high number concentrations at sizes 0.28 µm and 3 µm. Previous studies also measured lower number concentrations of small particles with an increase in large particles during several dust events (Ardon-Dryer and Levin, 2014; Niu et al., 2016).

An increase in particle number concentration during the dust event compared to the two haze events was observed for particles

≥ 0.8 µm. Observations based on the differences or ratio between PM$_{10}$ and PM$_{2.5}$ have been used to distinguish between dust and non-dust events (Alghamdi et al., 2015; Sugimoto et al., 2016). For the dust event, PM$_{10}$ - PM$_{2.5}$ was 277.6 µg m$^{-3}$, which

was an order of magnitude higher than in the two haze events (17 and 7.6 µg m$^{-3}$ for March 28 and 29, respectively). The PM$_{2.5}$/PM$_{10}$ ratio for the dust event was 0.15, while the values for the haze events were higher (0.61 and 0.74 for March 28 and 29, respectively). It has been suggested that a lower PM$_{2.5}$/PM$_{10}$ ratio (<0.35) indicates a contribution from natural sources
(e.g., dust event), while a higher ratio suggests a larger contribution from anthropogenic sources (Sugimoto et al., 2016; Tong et al., 2012). The PM$_{2.5}$/PM$_{10}$ ratio helps to distinguish between natural and anthropogenic events, but according to Lei and Wang (2014), this ratio may suffer from intrinsic deficiency as an identification criterion for dust events because the ratio for normal days may have already been very low. Therefore, additional measurements such as particle size distribution can support such observation.


As described above, continuous measurements of particle concentrations (number and mass) and particle size distribution enable distinguishing between dust and anthropogenic events in this area, which emphasizes the ability of AEROS's aerosol instruments to distinguish between different pollution events. The additional information of different PM sizes provided by AEROS, as well as total number concentrations and particle size distribution, can better explain the impact of different
pollution events on air quality in this region. Although the atmospheric measurements presented in this work were based on an hourly basis, each of the three instruments measures at a 1 min time resolution, allowing the observation of changes of particle concentration at short time intervals (e.g., 10 min). Measurements of such short duration will allow observation of short-term events that would have been missed when using the regular hourly average basis.

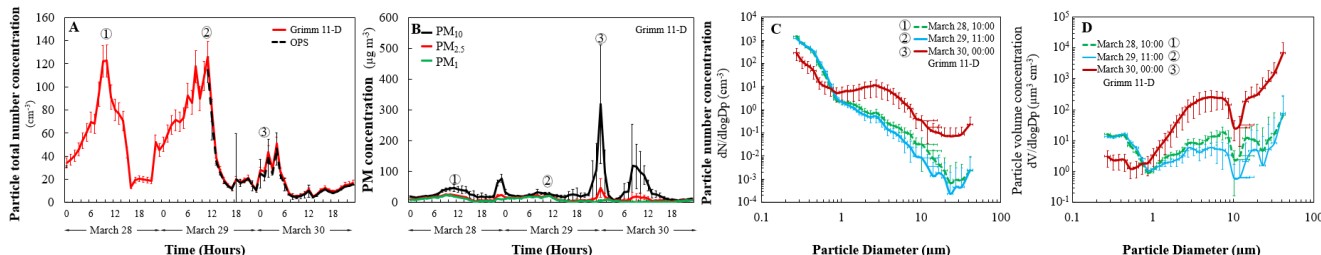


Figure 7. Measurements (hourly average) of total particle number concentration using OPS in black and Grimm 11-D, in red (A), measurements of PM mass concentration from Grimm 11-D (B), and particle number size distribution (C) and volume size distribution (D) of optical diameter using Grimm 11-D for March 28 - 30, 2019. The numbers on the plots represent different events (1 and 2 for the haze events and 3 for the dust event). Error bars represent SD values of hourly measurements.

**3.5. AEROS limitations**

Although AEROS provides crucial information on long-term measurements of various PM sizes, total particle number concentration, and particle size distribution under diverse meteorological and pollution conditions, it has some limitations. Some of these arise from the maintenance of AEROS, which requires weekly checks and calibrations, including cleaning of the instruments and inlets, and replacement of the silica gel in the dryers. The fact that all the instruments used are based on

optical size allows for comparison between the instruments, but also mean these instruments require examination and calibration by the manufacturer every year which could be a financial burden as the calibration cost for each unit can range from ~$3000 to ~$5000. While AEROS contain grammatic measurements for $PM_{2.5}$ and $PM_{10}$, those were not available at the time of this comparison and no access was available to reference units such as Beta Attenuation Mass (BAM) monitor or a Tapered Element Oscillating Microbalance (TEOM), therefore additional measurements under different atmospheric

conditions would be required to continue examination Grimm 11-D and DustTrak PM measurements. Another limitation is that our station provides information for only one site and is unable to capture the spatial variability of particles conversation, but even information from this one site is critical for this region, which does not have much information on atmospheric particles number concentrations, different PM sizes mass concentration or and particle size distribution.

## 4. Summary

The lack of AQMSs in the Southern High Plains inspired the design and building of AEROS, which provides continuous measurements of PM mass concentrations of various sizes, total particle number concentrations, and particle size distribution from three separate optical aerosol instruments (OPS, Grimm 11-D, and DustTrak). The three aerosol instruments provided overlapping measurements with similar mass and number concentrations of atmospheric and laboratory particles. Both the OPS and Grimm 11-D provided information on total number concentration and size distribution (at least for the size range of

$0.3 – 10\ \mu m$) and a comparison showed that they are very similar. The DustTrak and Grimm 11-D provided similar PM sizes; their comparison showed some differences depending on the PM sizes, although those differences were small, an additional examination will be required, ideally while using a reference PM measurement. Continuous measurement of particle concentrations and particle size distribution using AEROS allows demonstrating between dust and anthropogenic events demonstrating AEROS's ability to identify different pollution events, which will help us to better understand the impact of

diverse pollution events (mainly dust) on the air quality in this region.

**Author contribution.** KAD designed and built AEROS, designed the experiments, supervised the entire process, and performed most of the analysis, in addition to writing the manuscript. MK performed the experiments. XX helped with the data analysis. YD wrote all the MATLAB codes used for the analysis. All authors were actively involved in interpreting results
and in discussions on the manuscript.

**Declaration of competing interest.** The authors declare that they have no known competing financial interests or personal relationships that could have appeared to influence the work reported in this paper.

**Acknowledgment.** This research did not receive any specific grant from funding agencies in the public, commercial, or not-for-profit sectors. The authors would like to thank Livingston Jeff and Wilks Lee from National Wind Institute, and to James Browning and Darren Hedrick from the Department of Geosciences for all their help assembling AEROS. To Dr. Frank

Drewnick for his help and guidance, while using the Particle Loss Calculator (PLC), Texas Tech University for the support of Mary Kelley and Xia Xueting's scholarship and financial support.

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
