# Peer review of "The Aerosol Research Observation Station (AEROS)"

_Atmospheric Measurement Techniques, 2021_

## Author Comment (AC1)

Dear Editor,

Thank you for agreeing to consider a revision of our manuscript "The Aerosol Research Observation Station (AEROS)". We modified and revised the manuscript to address the reviewers' comments as well as to clarify points that they found confusing or unclear.

We would like to thank the two anonymous reviewers for their helpful comments and suggestions, and many thanks to you for your time and efforts with this revision. In line with the comments and suggestions, we revised the manuscript and made the requested additions and changes. Below are all the comments (in bold) followed by the replies. The parts that are in italic are corrections that are included in the revised version of the paper:

Sincerely,
Karin Ardon-Dryer

**Anonymous Referee #1**
**This manuscript describes a new aerosol monitoring station (AEROS) located in West Texas. Three mid-cost instruments (OPS 3330, Grimm-11D and DustTrak DRX) are installed, which provide PM mass concentration, particle number concentration and size distribution. Inter-comparisons and validation of the instruments in the laboratory as well as with atmospheric aerosols are presented. The manuscript is written in a clear language (apart from an issue with the units which is explained below) and the authors describe in adequate detail the monitoring site and facility.**

**I agree with the authors that mid-cost optical particle size spectrometers, such as the OPS 3330 and Grimm 11-D, seem to perform almost as good as high-end (and therefore more expensive) instruments with regards to particle number concentration and size distribution. I am not so convinced though about the performance of the Grimm 11-D and DustTrak DRX when it comes to PM mass concentration. As the authors correctly highlight, the algorithms used by these instruments are not disclosed. Moreover, calibration of PM monitors in the laboratory is not as standardised yet. In my opinion, mid-cost PM sensors are useful for providing high time-resolution data but there are still open questions about the accuracy of the results. Considering that PM mass concentration is the only regulated metric for aerosols (in most parts of the world), it is important that PM mass concentration is also monitored by the reference (manual) gravimetric method or, at least, by high-end instruments such as Beta Attenuation Mass (BAM) monitor or a Tapered Element Oscillating Microbalance (TEOM). In that sense, I think it is a pity that the manuscript does not provide any reference PM mass concentration data (the filter sampler unit was not operational). This could have increased the overall quality of the manuscript.**

We would like to thank the reviewer for the suggestions, corrections, and comments. We agree with the reviewer that a reference unit for PM monitor could have been ideal, but unfortunately such a unit (mainly because of its cost) was unavailable to us at the time and therefore could not be used as part of AEROS. Our filter gravimetric measurements were only available at a later stage, and we are currently working on this comparison.

Based on the reviewer suggestions we added a sentence about the reference PM units to the AEROS limitation section:

*The fact that all the instruments used are based on optical size allows for comparison between the instruments, but also mean these instruments require examination and calibration by the manufacturer every year which could be a financial burden as the calibration cost for each unit can range from ~$3000 to ~$5000. While AEROS contain grammatic measurements for $PM_{2.5}$ and $PM_{10}$, those were not available at the time of this comparison and no access was available to reference units such as Beta Attenuation Mass (BAM) monitor or a Tapered Element Oscillating Microbalance (TEOM), therefore additional measurements under different atmospheric conditions would be required to continue examination Grimm 11-D and DustTrak PM measurements. Another limitation is that our station provides information for only one site and is unable to capture the spatial variability of particles conversation, but even information from this one site is critical for this region, which does not have much information on atmospheric particles number concentrations, different PM sizes mass concentration or and particle size distribution.*

**Specific comments:**
- **I would like to kindly ask the authors to check carefully the units (cm-3 vs m-3) throughout the manuscript. Please make sure that number concentrations are given per cm3 and mass concentrations per m3.**

We would like to thank the reviewer for pointing our attention to this mistake, we apologize it happens. We checked and changed the units throughout the text and figures.

**Page 5/Line 148: According to the manual of the 11-D monitor, the number concentration can reach up to 3 000 000 particles/L (= 3 000 particles/cm3, not 3 000 000 particles/cm3 as stated in the manuscript) and mass concentration up to 100 mg/m3 (not 100 000 µg cm-3 as stated in the manuscript). Moreover, the manufacturer has recently revised the online technical specifications of the 11-D monitor to 5 300 000 particles/liter (https://www.grimm-aerosol.com/products-en/dust-monitors/the-dust-decoder/11-d/) (which we have also tested and confirmed in our laboratory).**

We would like to thank the reviewer for these corrections, changes were made to the revised manuscript:

*Data are recorded at 1 min intervals (it is also possible to save data every 6 s). Particle mass concentration can reach up to 100 mg m$^{-3}$, while number concentration can reach up to 53,000,000 # L$^{-1}$.*

**In Section 3.2 (text), mass concentrations are given in μg/cm-3 where it should have been μg/m-3. Sometimes, number concentration is expressed as #/cm-3 and some other times as #/cc (e.g. in Figure 4B). Please harmonise units throughout the text and figures.**

We would like to thank the reviewer for pointing our attention to this mistake, changes were made throughout the text and figures.

**Page 5, Lines 118 & 147: the unit of time is "s" (instead of "sec.").**

Changes were made according to the reviewer's suggestion.

**Page 3, Line 86: Consider revising "liters per min" to "L min-1" to be consistent with the rest of the manuscript.**

Changes were made according to the reviewer's suggestion.

- **Figures 3, 6, 7: When referring to particle diameter (x-axis), please specify what type of diameter this is (e.g. mobility, aerodynamic, geometric, optical etc.). In this case, I guess you are referring to optical diameters.**

We added information to the legend of each figure to reflect that the particle size distribution and diameter were based on the optical diameter.

*Figure 3. Comparison of the particle size distribution for optical diameter between the OPS and Grimm 11-D…*

- **Page 1, Line 21: More precisely, PMx is particulate matter suspended in air which is small enough to pass through a size-selective inlet with a 50 % efficiency cut-off at x μm aerodynamic diameter.**

This information was added to the revised manuscript:
*PM is categorized by the size of the particle, with PM$_{10}$ representing a mass of particles with an aerodynamic diameter up to 10 μm. PM$_4$, PM$_{2.5}$, and PM$_1$ representing a mass of particles with an aerodynamic diameter of up to 4, 2.5, and 1 μm, respectively. In general, PM measurements define as measurements where 50% of the particles with the defined diameter (e.g., PM$_{2.5}$) will pass through a size-selective inlet.*

- **Page 2/Line 54: Consider adding "… provided that they undergo a regular (e.g. yearly) service and recalibration".**

Information suggested by the reviewer was added to the manuscript.

The following information was added to the revised manuscript:
*Several studies have found mid-price optical particle sensors to be comparable to high-priced reference units as long as the mid-price optical particle sensors undergo a regular (e.g., yearly) service and re-calibration (Viana et al., 2015; Jaafari et al., 2018; Vasilatou et al., 2021).*

- **Throughout the text: The term aerosol concentration or particle concentration is vague. Please consider revising to "particles number concentration" or "particle mass concentration" as appropriate.**

Changes were made throughout the manuscript to reflect when mass or number concentration were used, we left the term particle concentration only when it was general and reflected both mass and number concentration.

An example of such from the revised manuscript:
*A comparison of each instrument pair (near each other) showed that both units measured similar overall concentrations (number and mass, data not shown).*

- **Page 3/Line 87: I would be interested to know whether the filters are conditioned (in Europe, filters must be conditioned in the weighing room at 19 °C to 21 °C and 45 % RH to 50 % RH for ≥ 48 h according to the standard EN 12341).**

Our filter room condition is based on EPA regulation, which is slightly different than the European one. Our filter room is kept in a room with the condition of temperature in the range of 20-23°C and RH of 30-40%.

We added information about this aspect to the revised manuscript:
*Filter are kept in the filter room which follows U.S. EPA (EPA, 1997) regulation conditions of temperature in the range of 20-23 °C and relative humidity in the range of 30-40%. To assure filter weight post sampling will not be impacted by hysteresis effect, filters post-aerosol collections are kept in a Dry-Keeper Auto-Desiccator Cabinet for 48 hours until weight.*

- **Page 5/Line 123: Do you mean "to convert the optical DIAMETER to aerodynamic sizes"?**

We thank the reviewer for pointing out this point, this is exactly what we wanted to write, we made changes in the sentence to make it clearer.

*… which are needed to convert the particle concentrations which are based on optical diameter to aerodynamic sizes.*

- **Page 14/Lines 360 & 376, Figure 7: How can the TOTAL particle number concentration be so low (122 # cm-3) when the number concentration of 0.25-0.3 μm particles is 10^5 # cm-3? Do you mean "total number concentration of particles with optical diameter larger than 1 μm"?**

We would like to thank the reviewer for this comment as it made us realize we plotted in Fig. 7C the raw count values and not the size distribution (dN/dlogDp) values. Corrections were made to Fig.7 and also into the text. Per suggestions from reviewer 2, we also change the figure to colored.

[Figure]

*Figure 7. Measurements (hourly average) of total particle number concentration using OPS in black and Grimm 11-D, in red (A), measurements of PM mass concentration from Grimm 11-D (B), and particle size distribution of optical diameter (C) using Grimm 11-D for March 28 - 30, 2019. The numbers on the plots represent different events (1 and 2 for the haze events and 3 for the dust event). Error bars represent SD values of hourly measurements.*

- **Page 16/Section 3.5: Please specify how often the instruments undergo maintenance and calibration at the manufacturer or another calibration laboratory. In my experience, light-scattering instruments need to be calibrated on a yearly basis.**

Information about maintenances and calibration was added to section 3.5 per the reviewer's suggestions.

*Some of these arise from the maintenance of AEROS, which requires weekly checks and calibrations, including cleaning of the instruments and inlets, and replacement of the silica gel in the dryers. The fact that all the instruments used are based on optical size allows for comparison between the instruments, but also mean these instruments require examination and calibration by the manufacturer every year which could be a financial burden as the calibration cost for each unit can range from ~$3000 to ~$5000.*

- **Figures 3 and 6: Please clarify in the caption what the error bars designate. Is it statistical uncertainties and at which confidence interval? Similarly, in Lines 273 and 281, please specify what the measurement uncertainties represent.**

Error bars represent standard deviation values from average for duration measurement represented by the time mentioned in the text. To clarify it we added information to the text and also to the figure caption.

*..the PM$_{10}$ particle mass concentration was 0.3 $\pm$ 0.16 $\mu g$ $m^{-3}$ (average $\pm$ standard deviation, SD values),*

The caption of figure 6 - *Figure 6. Comparison of measurements taken in the AEROS shed with measurements taken on the ground level or on the rooftop floor outside the AEROS shed. (A) Particle size distribution (optical diameter) measured by the Grimm 11-D unit in the AEROS shed (black) and outside on the rooftop floor (light gray). (B) Particle size distribution (optical diameter) measured by the OPS in AEROS (black) and on the ground floor (dark gray). (C) PM concentration at various sizes as measured by the Grimm 11-D unit in AEROS (black) and outside AEROS on the rooftop floor (gray). (D) PM concentration at various sizes as measured by DustTrak in AEROS (black) and on the ground floor (dark gray). Error bars represent SD values of the measured period.*

**Minor corrections:**
**Line 29: consider revising "to monitors…" to "to monitor…"**
Changes were made according to the reviewer's suggestion.

**Line 46: Consider adding "are", so that it reads "sensors are gaining popularity"**
Changes were made according to the reviewer's suggestion.

**Line 71: Consider adding a comma after "Texas" so that it reads "Lubbock, Texas, located…."**
Changes were made according to the reviewer's suggestion.

**Line 185: consider revising "There only reference…" to "The only reference…"**
Changes were made according to the reviewer's suggestion.

**Line 195: Nowadays, particles are usually made of polystyrene (without latex)**
We thank the reviewer for this correction, the word latex was deleted from the sentence.

**Line 199: Please provide the location of the company Powder Technology Inc.**

Changes were made, we added the location of the company.

*For this comparison, Arizona Test Dust (ATD) particles (Nominal 0 - 3 mm, Powder Technology Inc. MN, US)*

**References used in this document**

EPA (1997) Appendix L to part 50—reference method for the determination of fine particulate matter as PM2.5 in the atmosphere. Federal Register 62:57–95.

Jaafari, J., Naddafi, K., Yunesian, M., Nabizadeh, R., Hassanvand, M. S., Ghozikali, M. G., Nazmara, S., Shamsollahi, H. R., and Yaghmaeian, K.: Study of PM10, PM2.5, and PM1 levels in during dust storms and local air pollution events in urban and rural sites in Tehran, Hum. Ecol. Risk Assess.: Int. J., 24, 482-493, https://doi.org/10.1080/10807039.2017.1389608, 2018.

Vasilatou, K., Wälchlia, C., Koust, S., Horender, S., Iida, K., Sakurai, H., Schneider, F., Spielvogel, J., Wu, T. Y., and Auderset, K.: Calibration of optical particle size spectrometers against a primary standard: Counting efficiency profile of the TSI Model 3330 OPS and Grimm 11-D monitor in the particle size range from 300 nm to 10 µm, J. Aerosol Sci., 157, 105818, https://doi.org/10.1016/j.jaerosci.2021.105818, 2021.

Viana, M., Rivas, I., Reche, C., Fonseca, A. S., Pérez, N., Querol, X., Alastuey, A., Álvarez-Pedrerol, M., and Sunyer, J.: Field comparison of portable and stationary instruments for outdoor urban air exposure assessments, Atmos. Environ., 123, 220–228, http://dx.doi.org/10.1016/j.atmosenv.2015.10.076, 2015.

---

## Author Comment (AC3)

Dear Editor,

Thank you for agreeing to consider a revision of our manuscript "The Aerosol Research Observation Station (AEROS)". We modified and revised the manuscript to address the reviewers' comments as well as to clarify points that they found confusing or unclear.

We would like to thank the two anonymous reviewers for their helpful comments and suggestions, and many thanks to you for your time and efforts with this revision. In line with the comments and suggestions, we revised the manuscript and made the requested additions and changes. Below are all the comments (in bold) followed by the replies. The parts that are in italic are corrections that are included in the revised version of the paper:

Sincerely,
Karin Ardon-Dryer

**Anonymous Referee #2**
**Review of Ardon-Dryer et al., "The Aerosol Research Observation Station"**
**This manuscript provides a description and instrument performance evaluation for a new aerosol research station established in Lubbock, TX. The addition of these measurements is an important contribution to understanding aerosol sources and characterizing their physical properties in the region. Because Lubbock is in an area that is exposed to major dust events, these data will also help characterize dust properties of the region, measurements that are currently lacking. The authors did a nice job of organizing and describing the aerosol station and instrumental design. However, I have concerns and questions regarding the comparisons between instruments. Because of the large differences observed, it would be helpful if a reference PM2.5 and/or PM10 monitor could be incorporated as part of the instrumental design. I recommend publication after addressing these questions in the comments below.**

We would like to thank the reviewer for the suggestions, corrections, and comments. We agree with the reviewer that a reference unit for PM monitor would be ideal, but unfortunately, such a unit (mainly because of its cost) was unavailable to us at the time and therefore could not be used as part of AEROS. Our filter gravimetric measurements were only available at a later stage, and we are currently working on this comparison.

**Comments**
**Line 8: Include "mass" before "particles", as PM2.5 (or PM10) refers to the mass of particles with aerodynamic sizes less than a given size.**

Changes were made according to the reviewer's suggestion.

*...but only three AQMSs measure PM$_{2.5}$ concentrations (mass of particles with an aerodynamic diameter of <2.5 μm)*

**Line 11: include either "number" or "mass" concentration within "these particle concentrations". Mostly likely this would be "number" given the measurements.**

Changes were made according to the reviewer's suggestion. The following changes were implemented to the revised manuscript:

*The Aerosol Research Observation Station (AEROS) was designed to continuously measure these particles' (mass and number) concentrations using three optical particle sensors (Grimm 11-D, OPS, and DustTrak) to better understand the impact of dust events on local air quality.*

**Line 13: Consider providing additional information regarding the "three instruments used". Perhaps, 'the three optical particle sizers used'.**

The following changes were implemented to the revised manuscript per the reviewer recommendation:
*The Aerosol Research Observation Station (AEROS) was designed to continuously measure these particles' (mass and number) concentrations using three optical particle sensors (Grimm 11-D, OPS, and DustTrak) to better understand the impact of dust events on local air quality.*

**Line 15: Add either "mass" or "number" with "similar concentration measurement". Also, it would help to provide some quantifying information here besides "similar". Within a factor of 10?, typical biaes?, etc.**

Changes were made according to the suggestion; additional information about the comparison between the instrument was added to the abstract.

The following changes were implemented in the revised manuscript:
*This article provides a description of AEROS as well as an intercomparison of the different instruments using laboratory and atmospheric particles, which shows that the instruments used provided a similar range (within a factor of 3) of mass and number concentration measurements. Grimm 11-D and OPS show compatibility for comparison of number concentration and size distribution, and agreement in PM$_{10}$ concentrations (mass of particles with an aerodynamic diameter of <10 μm). Overall Grimm 11-D and DustTrak had a good agreement in mass concentration, comparison using laboratory particles was better than that with atmospheric particles. Overall, DustTrak measured lower mass concentrations compared to Grimm 11-D for larger particle sizes, and higher mass concentrations for lower PM sizes.*

**Line 17: It would also be helpful to indicate how these different episodes are distinguished-in this case by size distribution (not composition for example).**

Changes were made to the sentence to reflect the comment from the reviewer.

*Measurement with AEROS can distinguish between various pollution events (natural vs anthropogenic) based on their mass concentration and size distribution which will help to improve knowledge of the air quality in this region.*

**Line 20: Similar to earlier comment, include "mass concentrations of" after "representing" particles to indicate that PM10 data refer to mass concentrations.**

Changes were made to the sentence to reflect the comment from the reviewer.

*PM is categorized by the size of the particle, with $PM_{10}$ representing a mass of particles with an aerodynamic diameter up to 10 μm….*

**Line 35: Also, it might be complete to also include EPA's Chemical Speciation Network in this description as it is also a long-term US aerosol monitoring network, similar to IMPROVE but located in urban/suburban settings.**

Information about CSN was added to the revised manuscript as suggested by the reviewer.

*The EPA Chemical Speciation Monitoring Network (CSN) provides information on $PM_{2.5}$ and the chemical composition of ambient fine particles across 150 US urban sites (Solomon et al., 2014; EPA, 2022).*

**Line 46: Include "are" before "gaining"**

Changes were made according to the reviewer's suggestion.

**Line 103: Have any studies been performed to actually characterize the losses in the lines? Have losses been calculated based on theoretical calculations?**

Per the reviewer's suggestion, we perform particle loss calculation for the inlets (from rain protector to instrument). These calculations were made using the particle loss calculator presented in von der Weiden et al. (2009). Calculation loss was less than 5% for particles of 0.25 μm and less than 0.01% for particles in the size range of 1-2 μm. These values emphasize the efficiency of the inlets and the ability to collect particles with AEROS. This information was added to the revised manuscript.

*Calculation of particle loss in inlets (from rain protector to instrument) was performed using the particle loss calculator (von der Weiden et al., 2009). Calculations were made for particles in the size range of 0.25 to 41 µm (based on particle size measured by instrument), using different particle types of different density and shape factors (based values in Table 1 in Ardon-Dryer et al., 2015). Particle loss was below 5% for particles of 0.25 µm and below 0.01% for particles in the size range of 1-2 µm.*

**Line 119: Include "number" between "particle" and "concentration"**

Changes were made according to the reviewer's suggestion.

**Line 122: Some work in SW Texas might help inform as to the range of refractive indices and densities in the region (Hand and Kreidenweis, Aerosol Sci and Tech, 36, 1012-1026, 2002) during pollution and dust events.**

We thank the reviewer for pointing us to this paper, while the paper was very interesting, unfortunately, we cannot use the data from this paper as the two locations, Big Band national park examined by Hand and Kreidenweis (2002) paper and AEROS are very far from one another (~ 500 km, see figure below) and they are exposed to different dust source (perhaps different types/mineralogy of particles). As can be seen by winds rose of both location (for wind speed > 6 m s$^{-1}$; see figure below). Most of the dust and particles for AEROS come from the west, those from the south are mainly convective dust events that are small in scale and from nearby regions (Kelley and Ardon-Dryer, 2021). The Big Band dust origin mainly from the northeast. Since both site most likely does not have similar sources of particles it will be, to our opinion, a mistake to make an assumption they are similar, and therefore information of refractive indices and densities from this paper could be used. We are currently pursuing funding that will help us identify the chemical and mineralogy composition of particles in this region which will help us to understand the type of particles common which will help to identify their refractive indices and densities.

[Figure]

**Line 131: Note that here and several places in the paper the units are in error. Mass units are ug/m3 and number concentration are #/cc or #/cm3. I suggest looking closely at all instances of units in the paper to confirm. With respect to number concentrations, I suggest choosing one notation and keeping with it for the entire paper.**

We would like to thank the reviewer for pointing our attention to this mistake, we apologize for not being careful, changes were made throughout the text and figures.

**Line 183: Include "number" before "concentration"**

Changes were made according to the reviewer's suggestion.

**Line 195: What sizes of PSL are used for calibration? I think it is important to expand on these results if possible. The role of refractive index on reporting optical particle sizing data has been reported extensively in the literature and can significantly influence the results if a refractive index calibration isn't applied. Laboratory generated aerosol of known composition (ammonium sulfate, for example), in addition to Arizona test dust, can help provide uncertainties due to not accounting for varying refractive index.**

Information of comparison of OPS and Grimm 11-D using PSL particles was added to the revised manuscript and a new figure was added to the supplement part to show these measurements. As the reviewer stated previous studies examine the role of known particles (PSL and ammonium sulfate) on the performance of these two instruments under controlled laboratory settings. We do not believe that additional comparison using ammonium sulfate will be needed or in our case, it is not possible as we do not have an SMPS to size select the particles. We also believe that additional information (beyond what we added) on this matter will be beyond the scope of this paper, which focuses on the comparison and behavior of these instruments in the context of AEROS mainly under atmospheric settings. We believe the PSL comparison we added is sufficient enough to show the ability of the OPS and Grimm 11-D and the comparison of the two examined instruments.

The following were added to the revised manuscript:

*Although the three instruments were received from the manufacturer after factory calibration, we performed calibration tests of the OPS and Grimm 11-D using diverse monodisperse polystyrene sphere particles (0.25, 0.5, and 0.95 µm) to verify their performance in identifying particle size at the corrected size bins. The PSL particles were wet generated using a Brechtel Manufacturing, Inc. (BMI) 9200 Aerosol Generator (BMI, 2022), the atomized particles entered integrated in-line dryers where they evaporated, leaving anhydrous crystalline particles before reaching OPS and Grimm 11-D.*

*Analysis of OPS and Grimm 11-D using PSL particles was performed to identify if the instrument can detect particles at the correct sizes. Three different PSL sizes were examined, these PSL had nominal sizes of 0.25, 0.5, and 0.95 µm with a size range of 0.24-0.26, 0.48-0.52, and 0.93-0.97 nm, respectively. The results of the PSL test can be found in Fig. S2, on average 16 measurements were taken for each size and instrument. Overall, OPS and Grimm 11-D identified particles of similar sizes and concentrations. Both instruments, when examining 0.25 µm particles (Fig. S2A), had the highest concentration at the smallest (first) bin size, Grimm 11-D identified the 0.25 µm PSL at a bin size of 0.253 to 0.298 µm, while OPS detect the highest concentration at a bin size of 0.3 to 0.374 µm. When 0.5 µm PSL particles were examined (Fig. S2B), both units identified monodisperse distribution with a narrow maximum at the expected size range. The OPS identified the PSL at size range (bins) of 0.465 to 0.579 µm while Grimm 11-D identified most of the particles in two bins of 0.414 to 0.488 µm and 0.488 to 0.576 µm, the particles examined were in the range of 0.48-0.52 µm, and therefore identified in the correct detected sizes of Grimm 11-D. For the 0.95 µm particles (Fig. S2C), both instruments behave similarly and had bimodal distribution with two maxima, one at the smallest bin and another one at larger particle size. We suspected that this PSL solution was contaminated leading to this high concentration of small particles. OPS identified the 0.95 µm particles in size bins of 0.897-1.117 µm, while Grimm 11-D identified most of the particles in bin size of 0.679 to 0.8 µm, much lower than the PSL size range. More recently, when only Grimm 11-D was used (Fig. S2D) while using a new solution of 0.95 µm PSL particles, Grimm 11-D identified most of the particles in two bins 0.679 to 0.8 and 0.8 to 0.943 µm, the latter was in the PSL size range yet slightly lower than size expected. The detection of particles of that size range (~ 1 µm) at smaller sizes was observed in previous studies that used Grimm 11-D, yet it seems as if this size was in the detected size range according to ISO 21501-4 (Vasilatou et al., 2021). The behavior of the OPS came as no surprise as it was similar to previous studies that used size-selected ammonium sulfate particles (Ardon-Dryer et al., 2015).*

***Figure S2:*** *Comparison between OPS and Grimm 11-D using diverse sizes of PSL particles (0.25, 0.5, and 0.95 µm). Size distribution for each PSL tested for OPS (red) and Grimm 11-D (blue). Lines represent the average concentration over an average of 16 measurements and error bars represent standard deviation values of size bin and concentration measured.*

[Figure]

**Line 230: Here and throughout the manuscript, I have questions regarding how mass concentrations were calculated from the number concentration data. Was a constant density assumed? These instruments measure optical size, so converting the data to mass concentration would suggest the optical sizes were converted to aerodynamic sizes to get the PM size range reported? More details regarding these conversions is necessary.**

**It would also be helpful to show these comparisons in linear-linear plots, because some significant biases appear to be observed between these instruments, even for this known calibration aerosol. It might useful to provide slopes and intercepts so that the multiplicative and additive artifacts can be identified. A ratio of the total number concentration of one instrument to the other would be helpful because these log-log plots makes it difficult to assess performance between instruments. How were error bars determined on the size distribution plots? What errors do they incorporate? Why do they not appear symmetric?**

The manufacturer does not share information about the algorithm used to convert from optical size to mass, they also do not provide information on the refractive index, density, and weighting factors used for the calculations, not allowing the user to modify these values (with exception of OPS that allow changes of these values). The use of these instruments was as users; therefore, we examine their reading (mass concentration) and compare between instruments. We could not change parameters in the instrument or examine how these modify the mass concentration. We believe that such an examination would be beyond the scope of this work.

Per the reviewer's suggestions, we modify the figure and added more information on the comparison. We could not modify the figures to linear-linear plots as suggested by the reviewers as they looked weird, unclear to read, and could not allow the reader to observe the changes or comparison (see examples for figures below). In addition, it should be noted that previous studies present the size distribution ((dN/dlogDp) as a log-log plot.

Examples for figures showing linear-linear plots, for size distribution (left), and $PM_{10}$ (right).

[Figure]

As recommended by the reviewer, we delete the log figure comparison between the (mass and number) concentrations and added the values to a table presented in the supplementary section. We also added as suggested by the reviewer all the information of comparison including $R^2$, RMSE, MAE, slopes, intercepts, and also *P*-value for each comparison. We thank the reviewer for making this comment as we found that the two $R^2$ values that were plotted on the original figure were typed with a mistake, we corrected it checked all the values. Al the values are now presented in Table S1.

Per the reviewer comment and suggestion, we modify figure 3 and added a table with full detail to the supplement section:

*Arizona Test Dust particles were generated and measured by each instrument every minute for 30 min. A comparison of total particle number concentration and size distribution was made between the OPS and the Grimm 11-D, while a comparison of PM was performed between the DustTrak and Grimm 11-D. Overall, similar measurements were found between the various instruments as shown in Fig. 3. Full information on the statistics of each comparison including $R^2$, RMSE, and MAE, slope, intercepts, and the number of parallel measurements can be found in Table S1.*

[Figure]

*Figure 3. Comparison of the particle size distribution for optical diameter between the OPS and Grimm 11-D (A) total number concentration (B), and comparison of PM concentration between the DustTrak and Grimm 11-D for $PM_{10}$, $PM_4$, $PM_{2.5}$, and $PM_1$ (C) using Arizona Test Dust particles. Error bars represent SD values for measurement duration.*

*Table S1: Statistics for laboratory intercomparison of aerosol instrumentation using ATD particles*

| Variable compared | Instrument used | | AVE ± SD | | | | | | N | $R^2$ | RMSE | MAE | Slope | Intercepts | P values |
|---|---|---|---|---|---|---|---|---|---|---|---|---|---|---|---|
| Total Count | OPS | Grimm 11-D | 441 | ± | 210 | 505 | ± | 243 | 31 | 0.97 | 43 | 29 | 1.1 | 3.6 | 1.0 |
| $PM_{10}$ | Grimm 11-D | DustTrak | 9401 | ± | 20065 | 6050 | ± | 12866 | 33 | 1 | 721 | 378 | 0.6 | 32 | 0.9 |
| $PM_4$ | Grimm 11-D | DustTrak | 5092 | ± | 7983 | 2161 | ± | 5758 | 33 | 0.95 | 1214 | 7098 | 0.7 | -1427 | 0.9 |
| $PM_{2.5}$ | Grimm 11-D | DustTrak | 1904 | ± | 2325 | 1458 | ± | 5194 | 33 | 0.85 | 2005 | 11888 | 2.1 | -2455 | 1.0 |
| $PM_1$ | Grimm 11-D | DustTrak | 219 | ± | 296 | 1162 | ± | 5123 | 33 | 0.86 | 1894 | 1152 | 16 | -2352 | 1.0 |

*AVE ± SD - Average ± standard deviation, N - Number of parallel measurements (min), RMSE - Root-mean-square error, MAE - Mean absolute error and P values based on one-way ANOVA*

**Line 263: Again, for Figure 4a and Figure 5 for mass comparisons, what density was used to convert number size distribution data to mass concentrations and how were optical sizes converted to aerodynamic sizes for the PM1, PM4, etc. comparisons?**

Values of mass concentration are provided by each instrument and were not calculated by us; the manufacturer does not provide information on how these calculations were made from the density as we do not have that information or information on what density is used by Grimm 11-D and DustTrak. Users do not have access to that calculation or have the ability to modify these values (at least for the Grimm 11-D and DustTrak) and the manufacturer does not share information about it. We were using the instrument "as is" based on the manufacture calibration and did not convert the values from optical sizes to aerodynamic as we do not have information on particle type in this region and therefore on the expected refractive index or density.
We added information about the manufacture calibration to the information presented on each instrument.

*The OPS is calibrated by the manufacturer using different sizes of polystyrene latex sphere particles (PSL). In the operation of the OPS, the particle density is assumed as 1 g $cm^{-3}$, and no information on the reflective index is added, as there is very limited knowledge of the atmospheric particle chemical and mineralogical composition in this region…and, therefore, no way to correctly capture the particles' density or refractive index, which are needed to convert the particle concentrations which are based on optical diameter to aerodynamic sizes.*

*The DustTrak is calibrated by the manufacturer using Arizona Road Dust/ISO 12103-1, and the default calibration factor ("Factory Cal") of 1.0 was used (TSI Inc., 2019). No information is provided by the manufacturer on the calculation or measurements error of the DustTrak.*

*The signal from the scattered light is classified by size and count, and these counts are then converted to mass concentrations. These are made available through a Grimm proprietary algorithm, but the manufacturer does not share information about it, or the refractive index, density, and weighting factors used for the calculations. The Grimm 11-D is calibrated by the manufacturer using PSL particles according to ISO 21501-1, calibration factor ("Factory Cal") of 1.0 was used (Grimm 11-D, 2020).*

**Line 281: What two instruments are being compared with the two averages reported? Are these the AEROS instruments to the TCEQ?**

Changes were made to the sentence to clarify the comparison made

*In a comparison of PM$_{2.5}$ hourly values between the Grimm 11-D and DustTrak to the local TCEQ station (Figs. 5E, 5F) the AEROS instruments (Grimm 11-D and DustTrak) measured higher PM$_{2.5}$*

*values (with averages of 3.5 ± 5.5 and 6.1 ± 15.1 µg m⁻³, respectively) than those measured by the*
*TCEQ.*

**Line 294: Figure 5I shows number concentrations but average mass concentrations are listed here? Also, what is the experimental uncertainty of these instruments (are these differences within experimental uncertainty?)**

The values presented in Fig. 5I are the comparison of total number concentration as measured between the OPS and the Grimm 11-D. We are unsure about the confusion, regardless we made sure that the text, figure, and legend reflect that. We also separated between the section that covers the comparison of total number concentration and the summarize section.

The following changes were implemented in the revised manuscript:
*A comparison of total particle number concentration between the OPS and Grimm 11-D for particles 0.3 µm to 10 µm yielded a high $R^2$ value (0.98) and low RSME and MAE values (3.5 and 2.5 # cm⁻³, respectively), with a slope of 1.0 (Fig. 5I) emphasizing the compatibility of the two units.*

*Overall, the OPS and Grimm 11-D are more comparable based on their total number concentration and $PM_{10}$ values, but the Grimm 11-D and DustTrak had high comparison values (relatively high $R^2$ values)…*

Regarding the second part of the question, we are sorry we are unsure about the reviewer comment, regarding "experimental uncertainty of these instruments". If the question is regarding the range of values or quality of comparison the RMSE and MAE values who are based on each comparison can reflect that. If the reviewer means uncertainties of measurements of each measurement, only the OPS provides such information in the manual which was indicated in the method section. Unfortunately, that information is not provided for DustTrak manuals. We contacted the company to get such information. We received information about the Grimm 11-D and OPS and added it to the revised manuscript. Regarding the DustTrak, from the manual and communication with the manufacturer, we know that The DustTrak measures the mass resolution of 1 µg m⁻³, but based on TSI information on DustTrak accuracy isn't published or available (TSI, personal communication).

*..while number concentration can reach up to 53,000,000 # L⁻¹. The Grimm 11-D tolerance ranges are ± 3% for particle concentration ≥ 500 # cm⁻³, and ±2 µg m⁻³ (Grimm, personal communication).*

*The OPS time resolution is 1 min, with a flow rate is 1.0 L min⁻¹, which can reach a particle number concentration of up to 3,000 particle cm⁻³ with a size resolution of < 5% at 0.5 µm and with measurements error of 0.001 # cm⁻³ (TSI, personal communication).*

**Line 301: Can the authors provide more discussion about how particles may be "interpreted slightly differently" regarding these comparisons? It seems that understanding how and why the instruments are interpreting particles differently, especially since these are both optical measurements, would be important to understanding future measurements.**

Additional information was added to this section per the reviewer's comment. It's important to note that the manufacturer does not share information about the algorithm, or values of refractive index, density, and exact weighting factors calculation used for the calculations at lead not for DustTrak and Grimm so no changes could be made to reflect that. Since we do not have information on the type of particles (chemical composition and mineralogy) in this region we could not make any changes to the refractive index, the density of the OPS, which is the only unit out of the three that could be modified.

We added the following information and dissection to the revised manuscript as suggested by the reviewer:

*Some of these differences in mass concentration in the atmospheric measurements could be attributed to slight changes in the method used by each instrument for particle detection. For example, according to Wang et al. (2020), the OPS uses a more focused laser beam and a nozzle with a smaller inner diameter to sample particles compared to the one used in the DustTrak, while the DustTrak single scattering measurement has a larger minimum detectable size (~0.5 μm) yields more coincidence errors than the OPS. Another factor lay with the fact that the instruments are calibrated by the manufacturer using different particle types, both OPS and Grimm 11-D calibrated using PSL particles while the DustTrak is calibrated with Arizona Road Dust. Calibration using different particle types could cause different detection or reading. Previous studies indicated that optical responses of different particles may vary significantly, depending on the particles type or the pollution level (McNamara et al., 2011; Sousan et al., 2016; Masic et al., 2020). For example, irregular particles, like dust particles, will scatter more light which may overestimate the optical diameter of the particles (Chien et al., 2016). According to Zhang et al. (2018), the relationship between PM mass concentration and light scattering is strongly dependent on particle size and, to a lesser extent, on PM composition. Atmospheric particles, as the one used in this comparison, contain different types of particles which will be varied by their refractive indexes, densities, and shapes leading to slightly different interpretations by each of the instruments and to different readings (Cheng et al.,2010). Since there is very limited information about the atmospheric particle chemical and mineralogical composition in this region no correction (e.g., different refractive indexes, densities values) could be made, and instruments were used as default from the manufacture with manufacture correction factors.*

**Line 303: It would also be helpful to report intercepts from these linear regression as they are indicative of additive biases.**

Information of intercept was added to each of the comparisons as requested by the reviewer.

[Figure]

*Figure 5. Instrument comparison based on linear regression, comparison of hourly PM, and total particle number concentration values as measured by the Grimm 11-D, OPS, DustTrak, and TCEQ. Dashed gray lines represent a 1:1 line. The statistics of each case include the $R^2$, RMSE, and MAE, as well as the slope, intercepts (I), and N, which represent the number of parallel measurement points. Shown are comparisons of the Grimm 11-D and OPS (A) and Grimm 11-D and DustTrak (B) for $PM_{10}$ and between the OPS and DustTrak for $PM_{10}$ (C). The Grimm 11-D and DustTrak (A) and Grimm 11-D and TCEQ (B) for $PM_{2.5}$, and between TCEQ and DustTrak for $PM_{2.5}$ (E). Comparison between the Grimm 11-D and DustTrak for $PM_4$ (G) and $PM_1$ (H), and between Grimm 11-D and OPS for total particle number concentration (I).*

**Figure 303: Can the authors comment on the two different apparent subsets of data for the Figure I number concentrations? It appears 2 populations of data exist, one with fairly good agreement with 1:1 line and one with a multiplicative bias. Can the authors comment on why the shift? Was there a shift in calibration?**

We would like to thank the reviewer for the comment/question. We performed an in-depth analysis for the measurements of the total number of concentrations for each period, time between change

of silica gel in the dryer, with filter change in the instrument. We found two periods, in the middle of the measurements period, that had a larger difference between the two instruments resulting in the shift of the values from the 1:1 line. The cause for that is unclear, the period had an overall low PM concentration no big pollution event or change to the performance of the instrument. There was also no change of calibration during this entire measurement period. Information of this observation was added to the manuscript text and a figure was added to the supplemental section.

Information was added to the revised manuscript:
*It should be noted that although overall these two instruments show high comparability a close look at the distribution of the total concentration shows a difference between the OPS and Grimm 11-D over some periods. A comparison was performed between the units based on different periods, where each period represents the time between silica gel replacement and filter change in instruments (see Fig. S3). For two out of the nine periods (for unknown reasons) OPS measured much higher number concentration values compared to Grimm 11-D, leading to much higher difference values between the two units (Fig. S3B) and therefore shift of the 1:1 line (Fig. 5I).*

***Figure S3:*** *Comparison of total number concentration between OPS and Grimm 11-D during March-May 2019. Numbers of parallel measurements (hour) of total number concentration for both instruments per measurements period (A). Average and SD values (error bard) for the difference in number concentration between OPS and Grimm 11-D for each period, dash line highlight 0, no difference (B). Comparison between OPS and Grimm 11-D for total particle number concentration per period (different color), Dashed gray lines represent a 1:1 line.*

[Figure]

**Line 319: Can the authors expand on the intention for these additional comparisons? Were they testing the role of RH, height above ground? How can these various impacts be separated in the comparison? Was RH measured (or considered from met data) for the instruments with no RH control?**

Information on temperature and RH were retrieved from the meteorological (ASOS) station and not from instruments. Although the instrument provides information on temperature these usually

represent internal temperature, RH values are not provided. The goal of this sentence was to provide the reader information on the atmospheric conditions that this comparison was performed and to show that they were done on a similar range of conditions (small SD values).

Per the reviewer comments, we change the sentence so it will be clearer for the reader:
*These comparisons were taken under atmospheric conditions with a temperature of 26 ± 5.4 °C and relative humidity of 48.9 ± 16.7 % (as measured by the NWS station).*

**Line 324: Wouldn't these particles (kicked up from sidewalk) generally be larger than accumulation mode particles measured less than 0.5 um?**

We would like to thank the reviewer for this comment, as it made us realize we did not mention an important factor, which is the fact that ground measurements were taken close to an active parking lot (several meters), and it is highly likely there were some car activities at the time as the measurement itself took 1-hour long. We also believe we were not clear in our explanation as the higher concentration at the ground level was only for the OPS in size ranges from 0.3 to 2 µm. Based on that, we made corrections to the revised manuscript. It should be noted that previous studies found that walking can gerent particles at similar size ranges as we observed (e.g., Qian et al., 2014; Zhang and Yao, 2022).

We clarify this part and add more information in the revised manuscript:
*Overall, similar particle concentrations were found at all three locations (Fig. 6). The average particle size distribution measured in AEROS, when compared to those taken on the rooftop floor using the Grimm 11-D (Fig. 6A) showed similar number concentrations for all particle sizes. For the comparison between measured in AEROS and the ground floor using OPS (Fig. 6B), we found higher particle number concentration in size range of 0.3 to 2 µm (with difference up to 350 # cm$^{-3}$ for 0.3 µm) at the ground level. The measurements at the ground floor were higher most likely due to people walking near the instruments and kicking particles from the sidewalk, and the fact the ground sampling location was near a parking lot that was active during the sampling period.*

**Line 335: Can the authors expand regarding the "range of difference between the two instruments"? Is this with respect to the comparisons with ATD or experimental uncertainties?**

The sentence in line 335 describes the fact that the small differences were observed between the two DustTrak instruments (same type, ours and rental) when measurements were taken at the same location and time (as shown in the figure below). It is known that there are small differences between instruments (same type) therefore although we observed some differences between measurements that occur at the ground and the station they were in the same range as the differences found between the instruments when they were used at the same location and time.

This comparison was not performed with the ATD particles as we had the rental units for a very limited time. We modify the sentence to clarify it.

Figure showing average and SD values of mass concentration of two DustTrak measured in the same location and same duration (24h).

[Figure]

The following changes were made in the revised manuscript:
*Although there were differences in PM concentrations, these were relatively small (1.3 - 2.3 μg m⁻³) and within the range of difference found between the two instruments when they were measured at the same location and time.*

**Line 342: Can the authors comment regarding why the instruments outside the shed (Figure 6c) are consistently biased low for all size ranges? And the opposite is true for the Dusttrak at ground level? And can the authors comment on how error bars were calculated for these figure?**

We believe this comment is very similar to the previous one. The differences observed in figure 6C lay with the same explanation provided in the previous comment. There were small differences between the two instruments (same type), and the difference observed (in both Figures 6C and 6D) are in the same range found when the instrument was measured at the same location and time (see a figure in the previous comment). Regarding the error bars, these represent the standard deviation values of the average that was measured (over one hour, 60 measurements). We modify the sentence in the manuscript to clarify this point.

*Although there were differences in PM concentrations, these were relatively small (1.3 - 2.3 μg m⁻³) and within the range of difference found between the two instruments when they were measured at the same location and time.*

**Line 354: Is this site also influenced by biomass smoke?**

Yes, during late spring or early summer, there could be some biomass burning in this region, but they are much fewer compared to the dust event. No biomass smoke events occur during the measurements period present in this manuscript.

**Line 358: How was visibility assessed?**

Visibility values were retrieved from the NWS station; this information is provided in the Method section:

*Meteorological information, such as 5-min to hourly ambient temperature, relative humidity, wind speed, direction, and gust as well as visibility, pressure, and precipitation were retrieved from the local National Weather Service (NWS) Automated Surface Observation System (ASOS),..*

The following was added to the revised manuscript to clarify this question:
*The visibility (based on measurements taken from the meteorological station) decreased from 16 to 8 km…*

**Line 365: What did the TCEQ report for these periods?**

The TCEQ reported much lower values for these events compared to those measured by AEROS. We did not present the TCEQ values as part of this comparison in Figure 7 for two reasons, we did not want to overload the plot as we already present the Grimm 11-D values at three different PM sizes, in addiction TCEQ location is not near AEROS and therefore it experiences slightly different environmental conditions, as shown in the figure below. Overall TCEQ measure lower values than the Grimm unit but also its 0 calibrations allow it to measure negative PM concentration as shown in the figure for March 30. The different locations of AEROS and TCEQ units show that both experienced different conditions on 14:00 of March 28, where the TCEQ site was exposed to a local event that increase the $PM_{2.5}$ concentration but AEROS was not exposed to that event, therefore had a low $PM_{2.5}$ concentration. The dust event on March 30 is another example of the difference between the two based on their location. The strong wind originated from the east and passed by TCEQ before AEROS causing an increase of $PM_{2.5}$ concentration slightly earlier. We could not find satellite images to show this event and explain some of these differences between the two locations. We are currently working on an analysis of measurements from AEROS, we are planning to include more TCEQ data in that comparison as well as the composition of our gravimetric filter analysis from the Harvard impactor filter unit.

Figure showing the comparison of $PM_{2.5}$ hourly concentration from Grimm 11-D and TCEQ for March 28-30, 2019

[Figure]

**Line 369: Emissions will also largely influence the differences between haze events China and Texas.**

We agree with the reviewer about this point, it is expected that the rate of emission and perhaps also the type of emission will be different and will cause some of the differences we observed.

We added the following sentence into the revised manuscript to clarify this point:
*It is possible to assume that since measurements taken in this region which has much smaller cities compared to those measured in China, therefore there will be differences in the emissions rate and type which will attribute to the differences of number and mass concentrations observed here compared to those from China.*

**Line 405: Can the authors describe how error bars are calculated for number concentrations (figure 7c) and why they are not symmetric around the value?**

Error bars were calculated as the standard deviation (SD) values from the measurements averaged, for the duration examined. In the case of Fig. 7C, measurements were based on an hourly basis and therefore included 60 measurements (60 min), the SD values can vary based on the values measured during this hour. Information that describes the error bard was added to the manuscript and the description of the different figures presented with them.

*..the $PM_{10}$ particle mass concentration was $0.3 \pm 0.16$ µg m$^{-3}$ (average ± standard deviation, SD values),*

*Figure 7. Measurements (hourly average) of total particle number concentration using OPS in black and Grimm 11-D, in red (A), measurements of PM mass concentration from Grimm 11-D (B), and particle size distribution of optical diameter (C) using Grimm 11-D for March 28 - 30,*

*2019. The numbers on the plots represent different events (1 and 2 for the haze events and 3 for the dust event). Error bars represent SD values of hourly measurements.*

**Line 415: This adds to my confusion regarding how the authors converted their optical data to mass data?**

We apologize for the confusion; we modify the sentence and added recommended by reviewer 1.

*The fact that all the instruments used are based on optical size allows for comparison between the instruments, but also mean these instruments require examination and calibration by the manufacturer every year which could be a financial burden as the calibration cost for each unit can range from ~$3000 to ~$5000. While AEROS contain grammatic measurements for $PM_{2.5}$ and $PM_{10}$, those were not available at the time of this comparison and no access was available to reference units such as Beta Attenuation Mass (BAM) monitor or a Tapered Element Oscillating Microbalance (TEOM), therefore additional measurements under different atmospheric conditions would be required to continue examination Grimm 11-D and DustTrak PM measurements. Another limitation is that our station provides information for only one site and is unable to capture the spatial variability of particles conversation, but even information from this one site is critical for this region, which does not have much information on atmospheric particles number concentrations, different PM sizes mass concentration or and particle size distribution.*

**References:**
**I don't know how picky the journal will be, but some of the references don't include DOI's.**

Per the reviewer comment, we check our reference list and added DOI to every available journal we did not include on the first version of the manuscript. Some of the references were internet base links and therefore they do not have DOI numbers.

**Figures**
**Figure 1: The map quality is quite poor. The words on the Figure are hard to read. Define "NWS station", "TCEQ" and "HI" for reader who would have to flip back through the text. I assume the gray section of the tiny Texas map is the Southern High Plains? It would help to define. Also, label the instruments in the second photo.**

The figure was modified to reflect the review comment and suggestions. To reduce the text in the figure and make it simple we added identification letters for stations and numbers for instruments that now provided in the legend.

[Figure]

*Figure 1. Location of AEROS (A) in the South High Plains of West Texas with locations of meteorological station (B) and TCEQ PM$_{2.5}$ station (C). The photos show the filter sampler unit with Harvard Impactor (HI;1) units and the aerosol measurements unit (outside and inside view with dryers and instruments 2-Grimm 11-D, 3-OPS, and 4-DustTrak).*

**Figure 2: Define "ATD" and "PRIZE" in the caption.**

Definition for ATD and PRIZE was added to the figure caption and the manuscript text.

*Figure 2. Experimental setup: Arizona Test Dust (ATD) particles were generated using PRinted FluidIZed bed gEnerator- 3D dust generator (PRIZE) and measured by the various instruments (DustTrak, OPS, and Grimm 11-D).*

**Figure 4: These figures are very difficult to read. Can they be created in color?**

Figure 4 was changed to color as suggested by the reviewer.

[Figure]

[Figure]

*Figure 4. (A) Hourly values of PM₂.₅ from the Grimm 11-D (black), DustTrak (red), and TCEQ station (green) and (B) total number concentrations from the Grimm 11-D (black) and OPS (light blue) as measured during March-May 2019.*

**Figure 5: Typo for "slope" as "slop" in the figures.**

Changes were made according to the reviewer's suggestion.

[Figure]

**Figure 7: Can these figures be created in color? They are quite hard to read. Figure B: What are the instruments used in this figure?**

Information about the instrument used in Fig. 7B was added to the figure and the caption of the revised manuscript. We also changed the figure to color as suggested by the reviewer.

[revised manuscript text omitted]

Zhang, L and Yao M. Walking-induced exposure of biological particles simulated by a children robot with different shoes on public floors. Environment International, 158, 106935, https://doi.org/10.1016/j.envint.2021.106935, 2022.

---

## Author Response (AR2)

Dear Editor,

Thank you for agreeing to consider a revision of our manuscript "The Aerosol Research Observation Station (AEROS)". We modified and revised the manuscript to address the editor and reviewer 2 comments and suggestions. We would like to thank the anonymous reviewers and many thanks to you for your time and efforts with this revision.

In line with the comments and suggestions, we revised the manuscript and made the requested additions and changes. Below are all the comments (in bold) followed by the replies. The parts that are in italic are corrections that are included in the revised version of the paper:

Sincerely,
Karin Ardon-Dryer

**Comments to the author:**
**Thanks very much for your revisions in response to the reviewers, which I hope you agree have resulted in an improved manuscript. I'd like to see one more round of revisions, with further changes in response to the second round of comments. In addition, I have a few comments below that I'd like to see addressed. I will quickly review the revised manuscript without returning it to the reviewers.**

**Thanks for this nice contribution that highlights the need for improved coarse-mode on-line particle monitoring.**

**Line 16. What is meant by "provided a similar range (within a factor of three)"? Does this mean the instruments agree over their overlapping size ranges within a factor of three? Please be specific and quantitative.**

Baes on the editor's comment we rewrote these sentences to explain them better
*This article provides a description of AEROS as well as an intercomparison of the different instruments using laboratory and atmospheric particles. Instruments used in AEROS measured similar number concentration with an average difference of $2 \pm 3$ cm$^{-1}$ (OPS and Grimm 11-D using similar particle size ranges) and similar mass concentration, with an average difference of $8 \pm 3.6$ µg m$^{-3}$ for different PM sizes between the three instruments.*

**Line 17. What is meant by "show compatibility for comparison of number concentration and size distribution"? Compatibility could mean different things to different people. Please be specific with your terminology.**

We change the sentence to clarify it
*Grimm 11-D and OPS had similar number concentration and size distribution, using similar particle size range, and similar PM$_{10}$ concentrations (mass of particles with an aerodynamic diameter of <10 µm).*

**Line 12. Please state the size range covered by these three sensors.**

Changes were made to the sentence to reflect this comment.
*The Aerosol Research Observation Station (AEROS) was designed to continuously measure these particles' mass concentrations (PM$_1$, PM$_{2.5}$, PM$_4$, and PM$_{10}$) and number concentrations (0.25 – 35.15 μm) using three optical particle sensors (Grimm 11-D, OPS, and DustTrak) to better understand the impact of dust events on local air quality.*

**Line 71. The sentence beginning "Therefore" is a run-on with two "therefores" and needs to be split into two sentences.**

The sentence was split into two we also change the use of the word, therefore.
*Therefore, routine, and long-term measurements are required for comprehensive monitoring of diverse pollution events in this region, including dust events (Tong et al., 2012; Mahowald et al., 2014). Hence, there is a need to monitor particle mass concentrations (of various PM sizes) and size distribution to understand how they change under distinct metrological and pollution conditions.*

**Line 95. "filter substrates".**
Changes were made

**Line 110. Please make it clear that each instrument has its own separate inlet in tube. What is the total flow rate through each of the tubes?**

We added a sentence that will provide that information
*Each inlet is connected to a different instrument, the flow in each inlet varies based on the instrument used (1.0 or 1.2 L min$^{-1}$).*

**Line 170. Please use consistent units of cm^-3 instead mixing with L^-1. It's not common practice to use the "#" symbol for number, just say "cm^-3", and nomenclature consistency is important given the international readership of AMT.**

We apologize, we delete the use of L$^{-1}$ to define particle concentrations, we also remove the sign # from the text and the figures.

**Line 199. The dryers remove water from the particles by reducing the RH of the surrounding air. They do not impede hygroscopic growth.**

Changes were made to the sentence to reflect this comment
*The dryers remove water from the particles by reducing the relative humidity from the surrounding air, relative humidity after the dryer is low (24 ± 0.5%).*

**Line 2014. Change "measure" to "measures".**
We change the word "measure" to "measures".
**Line 220. "Diverse monodisperse" is an odd phrase. Perhaps "three monodisperse polystyrene sphere particle sizes"?**

Changes were made as suggested by the editor.

**Line 221-223. This is a run-on sentence. Please split into two sentences.**
Changes were made as suggested by the editor.

**Line 230. Change to "Brechtel".**
Changes were made as suggested by the editor.

**Lines 242-248. When there are uncertainties in both the x and y parameters, it is best to use a two-sided linear regression (e.g., orthogonal distance regression) to determine slope and intercept and associated errors. Were these regressions performed using standard least-squares linear regression? Is R^2 the square of the Pearson correlation coefficient? Would it be possible to repeat using ODR regression?**

The analysis in the paper was based on regular regression based on standard least-squares linear regression, we add that information to the paper manuscript. We also calculated the Pearson correlation coefficient and added that information to the supplement. We believe our use of linear regression was fine since the residual for both laboratory and atmospheric comparison (see fig below in orange) were of normal distribution. Regardless as suggested by the editor, we perform a new analysis based on the orthogonal distance regression (ODR) and added information of slop and Intercepts to each of the comparisons as part of the supplementary section.

[Figure]

Comparison of residuals of linear regression (orange; black line highlights the distribution) and orthogonal distance regression (blue) showing residual distribution for both comparison of ATD (upper panner) and atmospheric particles (lower panel).

The following information was added to the revised manuscript

*To evaluate the similarities and differences of the three instruments (or locations), a set of calculations and comparisons was performed using MATLAB and Excel. The evaluation and comparisons were based on R-squared ($R^2$), root-mean-square error (RMSE), and mean absolute error (MAE) values as well as the best fit information (including the slope and intercept), and Pearson correlation coefficient based on linear regression (standard least-squares linear regression). Additional evaluation based on orthogonal distance regression was made using R.*

*Full information on the statistics based on linear regression of each comparison including $R^2$, RMSE, and MAE, slope, intercepts, the number of parallel measurements, Pearson correlation*

*coefficient value as well as slop and intercepts based on orthogonal distance regression can be found in Table S1.*

*Additional information of each composition including averaged and SD, median, mode, 10th, and 90th percentile values can be found in Table S2.*

Table S1: Statistics for laboratory intercomparison of aerosol instrumentation using ATD particles based on linear regression

| Variable | Instrument used | | AVE ± SD | | N | Based on linear regression | | | | | | | Based on ODR | |
|---|---|---|---|---|---|---|---|---|---|---|---|---|---|---|
| | | | | | | $R^2$ | RMSE | MAE | Slope | Inter | PCC | P values | Slope | Inter |
| Total Count | OPS | Grimm 11-D | 441 ± 210 | 505 ± 243 | 31 | 0.97 | 43 | 29 | 1.1 | 3.6 | 0.984 | 1.0 | 1.2 | -6.2 |
| $PM_{10}$ | Grimm 11-D | DustTrak | 9401 ± 20065 | 6050 ± 12866 | 33 | 1 | 721 | 378 | 0.6 | 32 | 0.998 | 0.9 | 18.6 | -2926 |
| $PM_4$ | Grimm 11-D | DustTrak | 5092 ± 7983 | 2161 ± 5758 | 33 | 0.95 | 1214 | 7098 | 0.7 | -1427 | 0.977 | 0.9 | 0.7 | -1484 |
| $PM_{2.5}$ | Grimm 11-D | DustTrak | 1904 ± 2325 | 1458 ± 5194 | 33 | 0.85 | 2005 | 11888 | 2.1 | -2455 | 0.920 | 1.0 | 2.4 | -3044 |
| $PM_1$ | Grimm 11-D | DustTrak | 219 ± 296 | 1162 ± 5123 | 33 | 0.86 | 1894 | 1152 | 16 | -2352 | 0.927 | 1.0 | 0.6 | 25.8 |

AVE± SD - Average ± standard deviation, N - Number of parallel measurements (min), RMSE - Root-mean-square error, MAE - Mean absolute error, Inter- Intercepts, PCC- a value of Pearson correlation coefficient, P values based on one-way ANOVA and ODR-orthogonal distance regression. Total Count in cm$^{-3}$ and PM values in µg m$^{-3}$.

Table S2: Statistics of an intercomparison of AEROS instrumentation hourly measurements during March-May 2019

| Variable | Instrument used | | N | AVE ± SD | | | | Median | | Mode | | Standard Error | | 10th percentile values | | 90th percentile values | | PCC | ODR Slope | ODR Inter |
|---|---|---|---|---|---|---|---|---|---|---|---|---|---|---|---|---|---|---|---|---|
| Total Count | OPS | Grimm 11-D | 892 | 22.7 | ± 24.5 | 21.9 | ± 23.8 | 14.4 | 13.8 | NA | NA | 0.8 | 0.8 | 5.1 | 4.7 | 51.4 | 50.0 | 0.989 | 1.0 | 0.2 |
| $PM_{10}$ | OPS | Grimm 11-D | 867 | 11.5 | ± 14.9 | 20.3 | ± 23.8 | 7.8 | 14.9 | NA | 5.7 | 0.5 | 0.8 | 3.4 | 7.0 | 19.6 | 34.3 | 0.975 | 0.6 | -1.1 |
| $PM_{10}$ | OPS | DustTrak | 348 | 26.7 | ± 84.9 | 21.2 | ± 53.2 | 9.7 | 10.7 | NA | 6.7 | 4.6 | 2.9 | 4.3 | 6.0 | 26.6 | 23.0 | 0.889 | 0.6 | 5.3 |
| $PM_{10}$ | Grimm 11-D | DustTrak | 671 | 26.3 | ± 29.3 | 17.1 | ± 17.1 | 18.6 | 12.0 | 4.1 | 5.0 | 1.1 | 0.7 | 6.3 | 4.1 | 45.4 | 33.9 | 0.786 | 0.5 | 3.6 |
| $PM_4$ | Grimm 11-D | DustTrak | 671 | 17.1 | ± 15.9 | 14.3 | ± 14.8 | 13.3 | 9.2 | 7.5 | 2.0 | 0.6 | 0.6 | 4.9 | 3.3 | 29.6 | 29.8 | 0.853 | 0.9 | -1.4 |
| $PM_{2.5}$ | Grimm 11-D | DustTrak | 671 | 11.2 | ± 9.3 | 13.6 | ± 14.1 | 8.3 | 8.7 | 4.9 | 5.0 | 0.4 | 0.5 | 3.4 | 3.0 | 21.9 | 28.5 | 0.927 | 1.6 | -4.0 |
| $PM_1$ | Grimm 11-D | DustTrak | 671 | 7.7 | ± 6.9 | 13.0 | ± 13.4 | 4.8 | 8.3 | 1.5 | 2.0 | 0.3 | 0.5 | 1.7 | 3.0 | 17.1 | 27.4 | 0.863 | 2.1 | -3.4 |

N - Number of parallel measurements (min), AVE± SD - Average ± standard deviation, PCC- a value of Pearson correlation coefficient. Total Count in $cm^{-3}$ and PM values in $\mu g\ m^{-3}$. Slop and Inter (Intercepts) based on orthogonal distance regression (ODR).

**Line 268. The peak in the smallest bin when using the 0.95 µm PSL is probably due to the surfactant the manufacturer adds to the PSL mix. This surfactant helps keep the spheres from clumping together during storage, and often produce a tail of small particles. So I wouldn't describe this as "contamination", but "an artifact likely caused by surfactant within the PSL solution".**

We thank the editor for this information we were unaware of this issue, and we could not find such information on the PSL manufacture website. Per the editor's suggestion, we modify the sentence.

The following information was added to the revised manuscript:
*We suspected that high concentrations of small particles detected in this PSL solution were due to an artifact caused by the surfactant used in the PSL solution. The surfactant is added by the manufacturer to help keep the spheres PSL from clumping together during storage but often can produce a tail of small particles.*

**Line 287. Change to "($R^2$=0.97)". The minus sign is a little confusing.**
Changes were made as suggested by the editor.

**Lines 320-327. The standard deviations are larger than the mean values. This suggests that the probability distribution of the measurements is not normally distributed, and that means and standard deviations are not an appropriate way to statistically describe the data. If there is a long tail, a logarithmic transformation of the data might be appropriate. Alternatively, you could provide medians and 10th and 90th percentile values along with the mean and SD.**

The distribution measured was not normally distributed, the value represented differences between values measured by the different instruments (e.g., Grimm 11-D compared to DustTrak). Per the editor's comment, we added a new table to the supplementary section that contains additional information including Median, Mode, Standard Error, 10th, and 90$^{th}$ percentile values, PCC- a value of Pearson correlation coefficient, and Slop and Inter (Intercepts) based on orthogonal distance regression (ODR).

The following information was added to the revised manuscript:
*Additional information of each composition including averaged and SD, median, mode, 10th, and 90th percentile values can be found in Table S2.*

Table S2: Statistics of an intercomparison of AEROS instrumentation hourly measurements during March-May 2019

| Variable | Instrument used | | N | AVE ± SD | | | | | | Median | | Mode | | Standard Error | | 10th percentile values | | 90th percentile values | | PCC | ODR Slope | ODR Inter |
|---|---|---|---|---|---|---|---|---|---|---|---|---|---|---|---|---|---|---|---|---|---|---|
| Total Count | OPS | Grimm 11-D | 892 | 22.7 | ± | 24.5 | 21.9 | ± | 23.8 | 14.4 | 13.8 | NA | NA | 0.8 | 0.8 | 5.1 | 4.7 | 51.4 | 50.0 | 0.989 | 1.0 | 0.2 |
| $PM_{10}$ | OPS | Grimm 11-D | 867 | 11.5 | ± | 14.9 | 20.3 | ± | 23.8 | 7.8 | 14.9 | NA | 5.7 | 0.5 | 0.8 | 3.4 | 7.0 | 19.6 | 34.3 | 0.975 | 0.6 | -1.1 |
| $PM_{10}$ | OPS | DustTrak | 348 | 26.7 | ± | 84.9 | 21.2 | ± | 53.2 | 9.7 | 10.7 | NA | 6.7 | 4.6 | 2.9 | 4.3 | 6.0 | 26.6 | 23.0 | 0.889 | 0.6 | 5.3 |
| $PM_{10}$ | Grimm 11-D | DustTrak | 671 | 26.3 | ± | 29.3 | 17.1 | ± | 17.1 | 18.6 | 12.0 | 4.1 | 5.0 | 1.1 | 0.7 | 6.3 | 4.1 | 45.4 | 33.9 | 0.786 | 0.5 | 3.6 |
| $PM_4$ | Grimm 11-D | DustTrak | 671 | 17.1 | ± | 15.9 | 14.3 | ± | 14.8 | 13.3 | 9.2 | 7.5 | 2.0 | 0.6 | 0.6 | 4.9 | 3.3 | 29.6 | 29.8 | 0.853 | 0.9 | -1.4 |
| $PM_{2.5}$ | Grimm 11-D | DustTrak | 671 | 11.2 | ± | 9.3 | 13.6 | ± | 14.1 | 8.3 | 8.7 | 4.9 | 5.0 | 0.4 | 0.5 | 3.4 | 3.0 | 21.9 | 28.5 | 0.927 | 1.6 | -4.0 |
| $PM_1$ | Grimm 11-D | DustTrak | 671 | 7.7 | ± | 6.9 | 13.0 | ± | 13.4 | 4.8 | 8.3 | 1.5 | 2.0 | 0.3 | 0.5 | 1.7 | 3.0 | 17.1 | 27.4 | 0.863 | 2.1 | -3.4 |

N - Number of parallel measurements (min), AVE± SD - Average ± standard deviation, PCC- a value of Pearson correlation coefficient. Total Count in $cm^{-3}$ and PM values in $\mu g\ m^{-3}$. Slop and Inter (Intercepts) based on orthogonal distance regression (ODR).

**Fig. 5. "Concertation" typo in panel I.**

Changes were made as suggested by the editor.

**Fig. 7. It would be really interesting to see the number size distribution (panel C) converted to volume and plotted. There will be huge differences in the coarse mode (>1 μm), and volume is proportional to mass, which (in the form of PM10) is the main focus of this manuscript.**

Changes were made as suggested by the editor volume size distribution for each of the three events were added to figure 7

The following information was added to the revised manuscript:

[Figure]

*Figure 7. Measurements (hourly average) of total particle number concentration using OPS in black and Grimm 11-D, in red (A), measurements of PM mass concentration from Grimm 11-D (B), and particle number size distribution (C) and volume size distribution (D) of optical diameter using Grimm 11-D for March 28 - 30, 2019. The numbers on the plots represent different events (1 and 2 for the haze events and 3 for the dust event). Error bars represent SD values of hourly measurements.*

**Reviewer 2**
**The quality of the manuscript has improved but I still have a few questions and comments:**
**1) Line 170: The Grimm 11-D can measure number concentrations of up to 5 300 000**
**particles/L without coincidence losses (not 53 000 000 particles/L as stated in the text).**

We check again using the most updated Grimm 11-D manual below (Portable Aerosol
Spectrometer MODEL 11-D manual,2022), and the number concertation can reach 3.000.000 L$^{-1}$.
We believe the 5 300 000 particles/L we wrote based on the comment and suggestion from
reviewer 1comments was a mistake. The manual indicates the following *Number concentration 0*
*- 3.000.000 particle/Liter,* see image below. Reviewer 1 suggested this change based on
information from the brochure (see image below), but we check again, and the number was
changed in the manuscript accordingly to represent the correct number concertation up to 3,000
cm$^{-3}$.

Information from Manual   Number concentration   0 – 3.000.000 particel/Liter

| Particle number | 0 ... 5 300 000 particles/liter |
| --- | --- |

Information from brochure

The following information was added to the revised manuscript:
*while number concentration can reach up to 3,000 cm$^{-3}$.*

**2) Maybe I have missed something but I still do not understand Figure 7. Figure 7C shows**
**that the particle number concentration in the size bin 0.25-0.3 μm is as high as 1000 cm-3.**
**So how can it be that the total number concentration in Figure 7A is one to two orders of**
**magnitude lower? To calculate the total number concentration, shouldn't you calculate the**
**cumulative counts in all size bins and then divide by the sampled volume? In Figure 7A, the**
**concentration is sometimes as low as 5 cm-3. To achieve such low concentration in our lab**
**(clean-room conditions), we need to filter the air. I cannot see how the concentration can be**
**so low outdoors.**

The reviewer might think we are using small particle sizes. The smallest particle detected by our
instrument is 0.253 μm sizes. To explain the values measured from total count (Fig 7A) to
concentration (dN/dlogDp) in Fig 7C we provide the following explanation. The calculation for
dN/dlogD$_p$ is based on the eq. $dN/dlogDp = \frac{dN}{logD_{P,u} - ogD_{P,l}}$ where dN is the particle concentration
Dp is the midpoint particle diameter D$_{p,u}$ is the upper channel diameter, and D$_{p,l}$ is the lower channel
diameter. Therefore, if the particle concentration at the first size bin (0.253-0.298 μm) on March
28 at 10 am was 105.38 cm$^{-3}$ calculation of dN/dlogD$_p$ for this bin size will be $\frac{105.38}{log(0.298) - log(0.253)} =$
1482.3

We believe when the reviewer refers to a clean room, he is referring to different particle sizes. In
this case, the lowest particle number concentration was measured on March 30, with an hourly
total number concentration of 5.2 ± 0.5 cm$^{-3}$. This day was a Saturday, and we expected such low
concentration since very few cars normally enter the campus on weekend days. This area is
relatively clean and the measurements we receive show that. These concentrations will vary per
indoor/outdoor condition but most importantly it will be based on the particle sizes measured. We

believe that if we used instruments that detect smaller particles (e.g., SMPS, CPC) our TOTAL concentration would have been much higher.

**3) Depending on the meteorological conditions I suspect that the total particle number concentration can at times exceed the upper limit of the TSI OPS and Grimm 11-D (3'000 cm-3 and 5'300 cm-3, respectively).**

We check again the most updated Grimm 11-D manual, and the concertation can reach number concentration up to $3.000.000 \, L^{-1}$. It is possible to reach such a high concentration probably under very extreme pollution conditions, but from our experience so far, we have not experienced such behaviors. This area is rural and very clean overall (until we get a dust event). To emphasize it, in a recent work (Kelley et al., 2020) we analyzed the $PM_{2.5}$ values in this region over ~20 years and found that the daily values were below $10 \, \mu g \, m^{-3}$

**Have you considered adding a dilution unit upstream of these two instruments to avoid coincidence losses?**

We have been considering that option, it's a budget issue, but overall, we believe a dilution method is not a must for AEROS as we have not reached that high total concentration yet, and most air quality in our region is low until we get a dust event.

Reference used in this document
Portable Aerosol Spectrometer MODEL 11-D manual,2022
Kelley, M. C., Brown, M. M., Fedler, C. B., and Ardon-Dryer, K.: Long-term Measurements of PM2.5 Concentrations in Lubbock, Texas, Aerosol Air Qual. Res., 20, 1306-1318, https://doi.org/10.4209/aaqr.2019.09.0469, 2020.

---

## Author Response (AR3)

Dear Editor,

Thank you for agreeing to consider a revision of our manuscript "The Aerosol Research Observation Station (AEROS)". We modified and revised the manuscript to address your comments. We check the manuscripts for spelling and typographical errors, we also corrected the word concentration in Fig S3 as well as we checked the reference list, and make changes accordingly to AMT reference guidelines

Sincerely,
Karin Ardon-Dryer